# Benchmarking Missing Data Imputation Methods in Socioeconomic Surveys

**Siyi Sun**                                                   *siyi.sun@cs.ox.ac.uk*
*Department of Computer Science*
*University of Oxford*

**David Antony Selby**                            *david_antony.selby@dfki.de*
*German Research Center for Artificial Intelligence*

**Yunchuan Huang**                              *yunchuan.huang@eng.ox.ac.uk*
*Department of Engineering Science*
*University of Oxford*

**Ayush Patnaik**                                      *ayushpatnaik@gmail.com*
*XKDR Forum*

**Sebastian Vollmer**                              *sebastian.vollmer@dfki.de*
*German Research Center for Artificial Intelligence*

**Seth Flaxman**                                       *seth.flaxman@cs.ox.ac.uk*
*Department of Computer Science*
*University of Oxford*

**Anisoara Calinescu**                              *ani.calinescu@cs.ox.ac.uk*
*Department of Computer Science*
*University of Oxford*

**Reviewed on OpenReview:** *https://openreview.net/forum?id=HLhi9xhRw6*

## Abstract

Missing data imputation is a core challenge in socioeconomic surveys, where data is often longitudinal, hierarchical, high-dimensional, not independent and identically distributed, and missing under complex mechanisms. Socioeconomic datasets like the Consumer Pyramids Household Survey (CPHS)—the largest continuous household survey in India since 2014, covering 174,000 households—highlight the importance of robust imputation, which can reduce survey costs, preserve statistical power, and enable timely policy analysis. This paper systematically evaluates these methods under three missingness mechanisms: missing completely at random (MCAR), missing at random (MAR), and missing not at random (MNAR), across five missingness ratios ranging from 10% to 50%. We evaluate imputation performance on both continuous and categorical variables, assess the impact on downstream tasks, and compare the computational efficiency of each method. Our results indicate that

classical machine learning methods such as MissForest and HyperImpute remain strong baselines with favorable trade-offs between accuracy and efficiency, while deep learning methods perform better under complex missingness patterns and higher missingness ratios, but face scalability challenges. We ran experiments on CPHS and multiple synthetic survey datasets, and found consistent patterns across them. Our framework aims to provide a reliable benchmark for structured socioeconomic surveys, and addresses the critical gap in reproducible, domain-specific evaluation of imputation methods. The open-source code is provided in Appendix A.2.

## 1 Introduction

Missing data is a pervasive challenge in data science and machine learning, especially in real-world socioeconomic survey datasets (Silva-Ramírez et al., 2015; Wang et al., 2021). Data is often incomplete due to nonresponse or privacy concerns (Rubin, 2004). Imputation mitigates nonresponse bias and supports policy evaluation (Chen & Shao, 2000; Little & Rubin, 2019; Yang et al., 2024; Abdelnaby et al., 2024).

Despite the proliferation of imputation methods, there is a conspicuous lack of benchmarks to evaluate them on publicly available, large-scale, realistic datasets that capture the complexity of real-world socioeconomic survey data while allowing controlled introduction of missingness. Most empirical studies on missing data rely on relatively small datasets—such as the UCI machine learning repository (Zhang et al., 2025; Du et al., 2024; Miao et al., 2023; Bertsimas et al., 2018b) or limited clinical datasets (Zheng & Charoenphakdee, 2022)—or on synthetically generated data with simplistic assumptions (e.g., features drawn from a standard normal distribution) (Sun et al., 2023). Missingness is often simulated by randomly masking data under MCAR or MAR assumptions, which fail to reflect complex real-world patterns. In practice, missingness often follows the more challenging MNAR mechanism, where whether a value is missing depends directly on its unobserved value. Furthermore, numerous current benchmarks emphasize exclusively the accuracy of imputation, specifically evaluating the proximity of the imputed values to the actual values, while neglecting the consequences on downstream tasks (Zhang et al., 2025; Jarrett et al., 2022; Hastie et al., 2015; Biessmann et al., 2019). In practical applications, the goal of imputation is usually to enable reliable analysis or predictive modeling; thus, evaluating how different imputation methods affect the performance of subsequent tasks is crucial.

Although benchmarks exist for imputation on socioeconomic survey data (Wang et al., 2021; Li et al., 2024; Kalton & Kasprzyk, 1982), they typically suffer from several limitations, such as excluding MNAR scenarios, relying on a small set of missingness ratios (Bertsimas et al., 2018b), and lacking a systematic evaluation framework. To address these gaps, our work bridges the divide between restricted real-world data and reproducible experimentation by introducing a comprehensive and open benchmark for missing data imputation in socioeconomic surveys. To the best of our knowledge, we are the first to provide a large-scale, publicly shareable benchmark that integrates real, synthetic, and open socioeconomic datasets under diverse missingness scenarios and systematic evaluation metrics. Our contributions include the following:

- **Evaluations on Real, Synthetic, and Public Datasets:** We benchmark imputation methods on three datasets: the real-world CPHS (Pais & Rawal, 2021), its high-fidelity

synthetic counterpart SynthCPHS, and the publicly shareable SubSDIC derived from the World Bank's SDIC.

- **Comprehensive Missingness Scenarios:** We evaluate 14 imputation methods under three missingness mechanisms (MCAR, MAR, MNAR) and five missingness ratios, offering a broad and realistic spectrum of evaluation conditions.

- **Multi-metric Analysis & Downstream Task Evaluation:** In addition to the imputation accuracy on both continuous and categorical variables, we assess performance on downstream classification and regression tasks using multiple models to ensure robustness. We also systematically compare the computational efficiency of each method.

The remainder of this paper is organized as follows. Section 2 reviews related work on benchmark datasets, imputation methodologies, and synthetic data. Section 3 introduces the datasets used in our study. Section 4 defines the problem and missingness mechanisms. Section 5 describes our experimental setup and evaluation protocols. Section 6 presents results and analysis. Finally, Section 7 concludes the paper.

## 2 Related Work

### 2.1 Benchmark Datasets for Tabular Imputation

Research on imputation for tabular data often uses small, flat datasets like those from UCI machine learning repository (Kelly et al., 2025). As shown in Table 1, these datasets usually contain a few thousand to 100k samples with dozens of features, lacking clear hierarchical or temporal dependencies between variables. Details of these datasets are in Appendix A.4. Even more limiting, most studies simulate missing data, focusing on simplified MCAR or MAR scenarios with a single missingness level, which limits generalizability, since real-world data have more complex patterns.

A notable exception is the work of Jäger et al. (2021), who conducted a large-scale benchmark of imputation methods across 69 heterogeneous datasets from OpenML. Although their study offers valuable insight into general-purpose imputation performance, the datasets used are not drawn from the socioeconomic survey domain and lack the hierarchical and longitudinal structures typically present in national household surveys.

Recently, synthetic data benchmarks (Sun et al., 2023) have gained traction to test imputation algorithms under controlled conditions (e.g., varying missing rates or mechanisms); but most synthetic setups do not capture the complexity of real-world structured data. In particular, few, if any, existing benchmarks replicate the large-scale, multi-level characteristics of national socioeconomic surveys, which span millions of entries with state or regional hierarchies and repeated observations over time.

### 2.2 Imputation Methodologies

Approaches to imputing missing values can be grouped by their modeling philosophy. Statistical and iterative methods use repeated estimation cycles, including mean or mode filling, and classic algorithms like MICE (Van Buuren & Groothuis-Oudshoorn, 2011) and MissForest (Stekhoven & Bühlmann, 2012) that iteratively train predictors for each feature, as well as matrix-completion methods like SoftImpute (Hastie et al., 2015). These methods assume linear or low-rank structures

Table 1: Comparison of our proposed socioeconomic benchmarks against existing datasets commonly used in imputation literature (cited in $\geq 3$ studies). **Columns Definition:** *Samples*: Total number of rows; *Features*: Number of continuous ($N_{con}$) and categorical ($N_{cat}$) variables; *Evaluated Scenarios*: The number of different missingness ratios tested under MCAR, MAR, and MNAR mechanisms; *Capabilities*: Whether the benchmark supports Train/Test split evaluation, Downstream task evaluation, and contains hierarchical or longitudinal structures. Unlike standard UCI datasets (top), our proposed datasets (bottom) capture the complex, structured nature of real-world survey data.

| Dataset | Samples | Features | | Evaluated Scenarios ($N_{ratios}$) | | | Benchmark Capabilities | | | |
|---|---|---|---|---|---|---|---|---|---|---|
| | $(N)$ | $N_{con}$ | $N_{cat}$ | MCAR | MAR | MNAR | Train/Test | Downstream | Hierarchical | Longitudinal |
| *Existing General-Purpose Benchmarks (UCI)* | | | | | | | | | | |
| California Housing | 20k | 9 | - | 1 | 1 | 1 | ✓ | - | - | - |
| Letter Recognition | 20k | 16 | - | 1 | 1 | 1 | ✓ | - | - | - |
| Credit Card Clients | 30k | 14 | 9 | 1 | 1 | 1 | ✓ | - | - | - |
| Online News | 40k | 58 | - | 1 | 1 | 1 | ✓ | - | - | - |
| Concrete Strength | 1k | 8 | - | 4 | 4 | 4 | - | - | - | - |
| Wine Quality | 5k | 11 | - | 4 | 4 | 4 | - | - | - | - |
| Diabetes Health | 20k | 7 | 14 | 4 | 4 | 4 | - | - | - | - |
| SpamBase | 4k | 56 | - | 4 | 4 | 4 | - | - | - | - |
| *Proposed Socioeconomic Benchmarks* | | | | | | | | | | |
| **CPHS (Real)** | 1.4M | 16 | 8 | 5 | 5 | 5 | ✓ | ✓ | ✓ | ✓ |
| **SynthCPHS (Synthetic)** | 1M | 16 | 8 | 5 | 5 | 5 | ✓ | ✓ | ✓ | ✓ |
| **SubSDIC (Public)** | 500k | 6 | 12 | 5 | 5 | 5 | ✓ | ✓ | ✓ | - |

and often struggle with complex nonlinear feature interactions. Recent methods such as MIRACLE (Kyono et al., 2021) introduce a causally aware regularization that models the missingness mechanism jointly with the data, encouraging imputations consistent with the underlying causal structure. Distribution-matching methods align observed and imputed distributions: MOT (Missing-data Optimal Transport) (Muzellec et al., 2020) formulates imputation as finding the allocation of missing values that minimizes the optimal transport distance between batches of incomplete data, while its successor TDM (Transformed Distribution Matching) (Zhao et al., 2023) learns a nonlinear mapping before applying optimal transport to better capture the data's intrinsic geometry. These methods achieve state-of-the-art accuracy on many benchmark tasks and are particularly effective for in-sample imputation but generalize poorly to new records since they treat missing entries as learned model parameters.

Deep generative models (VAE (Mattei & Frellsen, 2019), GAN (Yoon et al., 2018), diffusion (Zheng & Charoenphakdee, 2022)) capture joint distributions of observed and missing data. While these models can capture complex nonlinear dependencies, they often face challenges in estimating distributions from incomplete data and in performing conditional inference, especially under high missingness. To overcome these issues, recent methods combine generation with iterative refinement. For example, **DiffPuter** (Zhang et al., 2025) integrates diffusion models into an EM framework, using iterative E- and M-steps to improve imputation quality.

Hybrid deep learning methods blends machine learning pipelines with automated model selection or specialized architectures. **HyperImpute** (Jarrett et al., 2022) employs an AutoML-style pipeline that selects the best model for each variable and updates imputations iteratively. Other architectures leverage advanced designs: **DSAN** (Lee & Kim, 2023) applies self-attention to learn feature and sample dependencies via masked reconstruction, while **ReMasker** (Du et al., 2024) extends masked autoencoding by re-masking observed entries during training, promoting robustness across different missingness patterns. Deep learning models have demonstrated robust performance on challenging

imputation tasks, especially when missing rates are high or feature types are heterogeneous (Zhang et al., 2025).

Some recent studies have empirically evaluated the "impute-then-predict" pipeline. For example, Bertsimas et al. (2018a) and Poulos & Valle (2018) proposed frameworks that highlight the importance of integrating imputation with supervised learning tasks. More recent work by Paterakis et al. (2024) questions whether explicit imputation is always necessary in predictive pipelines, particularly within the context of AutoML. However, our work places equal emphasis on two complementary objectives: restoring incomplete datasets to reduce the need for costly follow-up surveys, and ensuring robust performance on downstream tasks.

### 2.3   Synthetic Data for Imputation

When real-world socioeconomic data is private or lacks ground truth, synthetic data provides a practical alternative for benchmarking imputation methods. Prior work has used synthetic simulations to evaluate methods under controlled conditions (Kyono et al., 2021; Sun et al., 2023), but most rely on simple i.i.d. data or low-dimensional toy settings (Muzellec et al., 2020; Bertsimas et al., 2024), lacking the structural and statistical complexity of real surveys. National household surveys like CPHS feature multi-level hierarchies, repeated measurements, and non-random missingness, but their proprietary nature limits open evaluation. To address this, we introduce a synthetic benchmark dataset, **SynthCPHS**, which we design to replicate the structure and distribution of CPHS. We validate the similarity using the Kolmogorov–Smirnov (KS) test and Jensen–Shannon (JS) divergence, and the synthetic construction enables GPU-supported evaluation beyond CPHS's secure CPU-only server environment. We also present **SubSDIC**, a public subset derived from World Bank data (World Bank, 2023), which supports systematic and reproducible benchmarking in the socioeconomic domain.

## 3   Datasets

### 3.1   CPHS & The SynthCPHS Dataset

#### 3.1.1   CPHS

The Consumer Pyramids Household Survey (CPHS) by CMIE is the largest continuous household survey in India, running since 2014 and covering a panel of over 174,000 sample houses (about 111,000 rural and 63,400 urban) spread across most states in India surveyed thrice yearly (Pais & Rawal, 2021; Somanchi, 2021). It captures a broad array of household attributes including labor supply, income, consumption (expenditure on various needs), borrowing, and asset ownership (Pais & Rawal, 2021). This breadth makes it highly valuable for socioeconomic analysis (Chatterjee & Dev, 2023; Kathuria & Dev, 2024; Jagannarayan & Prasuna, 2024). In this paper, we select 25 socioeconomic features, as shown in Figure 1). Because some variables were missing in earlier waves, we restrict the analysis to complete cases from waves 18–30 (each wave is a survey round), preserving the longitudinal and hierarchical structure and yielding 1,341,651 records.

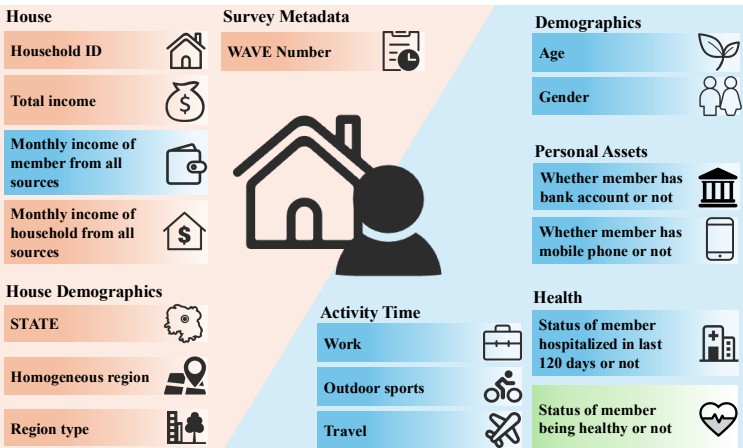

Figure 1: Overview of selected features in the CPHS and SynthCPHS datasets. Household-level (orange) and individual-level (blue) features form a hierarchical structure. The target variable (green) indicates individual health status for downstream classification. The wave number indexes each survey round, capturing longitudinal structure.

### 3.1.2  SynthCPHS

As CPHS is proprietary, available only by subscription, and restricted to a secure CPU-only server, benchmarking is severely constrained, especially for GPU-based imputation. To address this, we construct SynthCPHS, a synthetic dataset that mirrors the statistical properties and structure of CPHS but can be used on external GPU systems, enabling full-scale evaluation of imputation accuracy and efficiency. SynthCPHS includes 1,000,000 records and shares the same feature sets with the CPHS dataset as shown in Figure 1. The dataset was generated using the `synthpop` package in R (Nowok et al., 2016), a widely used tool to produce artificial microdata that preserve the statistical properties of the original survey while protecting individual confidentiality (Nowok et al., 2017). To support its validity, we compared the marginal and joint distributions of key features in SynthCPHS and CPHS using the Kolmogorov–Smirnov (KS) test and Jensen–Shannon (JS) divergence, finding no significant distributional differences.

### 3.2  SubSDIC

To ensure reproducibility, we introduce **SubSDIC**, a public subset dataset derived from the World Bank's Synthetic Data for an Imaginary Country (SDIC)-a fully synthetic census dataset representing an imaginary middle-income country. SDIC was generated using REaLTabFormer (Solatorio & Dupriez, 2023), a deep generative model trained on global household survey data, including IPUMS International, DHS, and the World Bank Global Consumption Database. SDIC consists of two flat tables for household- and individual-level attributes. We join these tables via household ID, select 19 mixed-type variables spanning both levels, and randomly sample 500k records from the full 10 million to construct SubSDIC. We designate the individual's highest educational attainment (`cat_educ_attain`) as the target variable for downstream classification and years of schooling (`con_yrs_school`) as the target variable for downstream regression. SubSDIC preserves realistic socioeconomic frameworks, including hierarchical relationships among households, individuals,

provinces, and districts, allowing for regulated evaluation of imputation accuracy, efficiency, and downstream performance across different missing data scenarios. Kolmogorov-Smirnov tests and Jensen-Shannon divergence analysis confirm that SubSDIC closely mimics the marginal distributions of the original SDIC dataset. Detailed feature descriptions are provided in Appendix A.5.

## 4 Problem Definition and Missing Mechanism

### 4.1 Problem Definition

Let $\mathbf{X}$ denote the matrix $n \times d$ that contains the complete data values in the variables $d$ for all $n$ units in the sample. Define the mask variable $\mathbf{M}$ as an $n \times d$ 0-1 matrix indicating whether a data point of $\mathbf{X}$ is observed (1) or missing (0). The elements of $\mathbf{X}$ and $\mathbf{M}$ are denoted by $x_{ij}$ and $m_{ij}$, respectively, where $i = 1, ..., n$ and $j = 1, ..., d$. We further define the partially observed data matrix as $\widetilde{\mathbf{X}}$, and its elements $\tilde{x}_{ij}$, such that

$$\tilde{x}_{ij} = \left\{ \begin{array}{lll} x_{ij}, & \text{if} & m_{ij} = 1 \\ \emptyset, & \text{if} & m_{ij} = 0 \end{array} \right. ,$$

Here $\emptyset$ represents an unobserved value. In the missing data imputation problem, the task is imputing data matrix $\widehat{\mathbf{X}}$ from the observed data matrix $\widetilde{\mathbf{X}}$ and make it as similar as possible to the complete data matrix $\mathbf{X}$.

### 4.2 Missingness Mechanism

Little & Rubin (2019) find it helpful to differentiate between the missingness mechanisms, which refers to the relationship between the occurrence of missing data and the values of the variables in the data matrix. The missing mechanism indicates whether the occurrence of missingness is connected to the underlying values of the variables in the dataset. The importance of missingness mechanisms lies in the fact that the effectiveness of data imputation methods is highly influenced by the specific dependencies present in these mechanisms. Therefore, we introduce the three missingness mechanisms as defined by Rubin (1976) here. Let $\mathbf{X}$ and $\mathbf{M}$ be defined as in Section 4.1. Assume the rows $(\mathbf{x}_i, \mathbf{m}_i)$ are i.i.d. across $i$. The missingness mechanism is specified by the conditional distribution $p_{\mathbf{M}|\mathbf{X}}(\mathbf{m}_i \mid \mathbf{x}_i, \boldsymbol{\theta})$, where $\boldsymbol{\theta}$ denotes unknown parameters.

- **MCAR**: If the missingness is independent of the data values, either missing or observed, this means for all $i$ and any distinct values $\mathbf{x}_i, \mathbf{x}_i^*$ in the sample space of $\mathbf{X}$, the conditional distributions are equal:

$$p_{\mathbf{M}|\mathbf{X}}(\mathbf{m}_i \mid \mathbf{x}_i, \boldsymbol{\theta}) = p_{\mathbf{M}|\mathbf{X}}(\mathbf{m}_i \mid \mathbf{x}_i^*, \boldsymbol{\theta})$$

  where $\mathbf{x}_i^*$ serves as a placeholder for a distinct, hypothetical value that could take the place of $\mathbf{x}_i$ in the sample space of $\mathbf{X}$.

- **MAR**: Let $\mathbf{x}_{(1)i}$ represent the observed components of $\mathbf{x}_i$ and $\mathbf{x}_{(0)i}$ represent the missing components of $\mathbf{x}_i$. A less restrictive assumption than MCAR is that the missingness depends on $\mathbf{x}_i$ only through the observed components $\mathbf{x}_{(1)i}$. This implies that for any distinct values $(\mathbf{x}_{(0)i}, \mathbf{x}_{(0)i}^*)$ of the missing components within the sample space of $\mathbf{x}_{(0)i}$, the probability of

missingness remains the same. In mathematical terms, the conditional distribution of the missingness mechanism can be expressed as:

$$p_{\mathbf{M}|\mathbf{X}}(\mathbf{m}_i \mid \mathbf{x}_{(0)i}, \mathbf{x}_{(1)i}, \boldsymbol{\theta}) = p_{\mathbf{M}|\mathbf{X}}(\mathbf{m}_i \mid \mathbf{x}_{(0)i}^*, \mathbf{x}_{(1)i}, \boldsymbol{\theta}) \qquad (1)$$

- **MNAR**: Unlike the MAR mechanism, where missingness is related to the observed values, if missingness is dependent on the unobserved values, the mechanism is classified as MNAR. The distribution of $\mathbf{m}_i$ depends on the missing components of $\mathbf{x}_i$, which means that equation 1 is not valid for some units $i$ and some values $(\mathbf{x}_{(0)i}, \mathbf{x}_{(0)j}^*)$ of the missing components.

### 4.3 Missingness Generation

In this study, missing values are synthetically introduced into complete datasets to establish ground-truth benchmarks. We adapt procedures from the R-miss-tastic platform (Mayer et al., 2019) to generate missingness under MCAR, MAR, and MNAR mechanisms. Crucially, the generation process is fully independent for every experimental configuration; a new missingness mask is generated for each sample dataset.

**Missing Completely at Random (MCAR)**  Missingness is independent of observed and unobserved data. For each feature $j$ and sample $i$, the missingness indicator $m_{ij}$ follows a Bernoulli distribution $P(m_{ij} = 1) = \alpha$, where $\alpha$ is the target proportion.

**Missing at Random (MAR)**  Missingness in feature $\mathbf{X}_j$ depends on other observed features but not on $\mathbf{X}_j$ itself. We employ a feature-wise logistic model as outlined in **Algorithm 1**. To strictly adhere to the MAR assumption, the weight matrix $\mathbf{W}$ is constructed with a zero diagonal ($\mathbf{W}_{j,j} = 0$). The influencing covariates are assigned uniform weights $\beta_k = 1/(d-1)$ for all $k \neq j$ in a $d$-dimensional dataset, ensuring bias-free dependency on the remaining features.

**Missing Not at Random (MNAR)**  MNAR allows missingness to depend on the feature's own value. As shown in **Algorithm 1**, this is achieved by initializing $\mathbf{W}$ as an all-ones matrix (retaining $\mathbf{W}_{j,j} = 1$) and normalizing the weights row-wise to $\beta_k = 1/d$, thereby allowing the missingness of a variable to depend on all features, including itself.

**Distributional Bias and Extrapolation.**  The logistic missingness generation mechanism (Algorithm 1) inherently introduces *distributional bias* by assigning higher missingness probabilities to extreme values. This simulates **tail censoring**, aligning with empirical findings in survey methodology where high- and low-income households are significantly less likely to disclose earnings (Riphahn & Serfling, 2005; Meyer et al., 2015).

## 5 Experimental Setup and Evaluation

### 5.1 Dataset Distribution Comparison

We evaluate how closely the synthetic dataset (SynthCPHS) reproduces the marginal distributions of the real CPHS across continuous and categorical variables.

---

**Algorithm 1:** Generation of Missing Values under MAR/MNAR Mechanism

---

**Input** : Complete Matrix $\mathbf{X}$ ($n \times d$), Missingness Ratio $\alpha$, Mechanism="MAR" or "MNAR"

**Output**: Partially Observed Data Matrix $\widetilde{\mathbf{X}}$

// Initialization

1 $\widetilde{\mathbf{X}} \leftarrow \mathbf{X};\quad \mathbf{W}^{d \times d} \leftarrow \mathbf{1}\mathbf{1}^\top$ ;                          // Initialize `d × d` matrix with ones

// Step 1: Define Dependency Structure

2 **if** *Mechanism=="MAR"* **then**

3 $\quad|\quad \mathbf{W} \leftarrow \mathbf{W} - \mathbf{I}_d$ ;                          // Remove self-dependency

4 $\mathbf{W} \leftarrow \text{Normalize}(\mathbf{W}, \text{axis} = \text{row})$ ;                          // ensure $\sum_k \mathbf{W}_{j,k} = 1$

// Step 2: Generate Missingness

5 $\mathbf{S} \leftarrow \mathbf{X}\mathbf{W}^\top$ ;                          // Compute weighted scores `(n × d)`

6 $\mathbf{Z} \leftarrow \text{Normalize}(\mathbf{S}, \text{axis} = \text{col})$ ;                          // Standardize scores feature-wise

7 Determine vector $\boldsymbol{\gamma} \in \mathbb{R}^d$ such that $\text{mean}(\sigma(\mathbf{Z} + \boldsymbol{\gamma}), \text{axis} = \text{col}) \approx \alpha$;

8 $\mathbf{P} \leftarrow \sigma(\mathbf{Z} + \boldsymbol{\gamma})$ ;                          // Compute probability matrix `(n × d)`

9 Generate $\mathbf{M}$ from Bernoulli distribution with parameter $\mathbf{P}$ ; // Mask matrix (1 = Missing)

// Step 3: Post-processing Safety Check

10 Find indices $\mathcal{I} \leftarrow \{i \mid \sum_{j=1}^d \mathbf{M}_{i,j} = d\}$ ;                          // Find rows that are fully missing

11 **if** $\mathcal{I}$ *is not empty* **then**

12 $\quad|\quad$ Sample indices $\mathbf{k} \in \{1, \ldots, d\}^{|\mathcal{I}|}$ uniformly;

13 $\quad|\quad \mathbf{M}[\mathcal{I}, \mathbf{k}] \leftarrow 0$ ;                          // Ensure at least one observed value per row

14 $\widetilde{\mathbf{X}}[\mathbf{M}] \leftarrow \text{NA}$ ;                          // Apply final mask to data

15 **return** $\widetilde{\mathbf{X}}$

---

- **Method for continuous variables: two-sample KS test.** Let $\{x_i\}_{i=1}^n$ and $\{y_j\}_{j=1}^m$ be two i.i.d. samples with empirical CDFs $F_n$ and $G_m$. The two-sample Kolmogorov–Smirnov statistic is

$$D_{n,m} = \sup_{x \in \mathbb{R}} \left| F_n(x) - G_m(x) \right|. \tag{2}$$

Under the null hypothesis that both samples come from the same continuous distribution, $\sqrt{\frac{nm}{n+m}} D_{n,m}$ converges in distribution to the Kolmogorov distribution, which yields exact or asymptotic $p$-values (Massey Jr, 1951). In all our results we report the unscaled statistic $D_{n,m}$ in Eq.(2), denoted as "KS" in the figures, together with its two-sided $p$-value.

- **Method for categorical variables: JS divergence.** For two discrete distributions $P$ and $Q$ over the same support $\mathcal{X}$, the JS divergence is the symmetrised, smoothed version of the Kullback-Leibler (KL) divergence:

$$\text{JS}(P\|Q) = \tfrac{1}{2}\text{KL}\left(P\|M\right) + \tfrac{1}{2}\text{KL}\left(Q\|M\right), M = \tfrac{1}{2}\left(P + Q\right), \tag{3}$$

where $\text{KL}(P\|Q) = \sum_{x \in \mathcal{X}} P(x) \log \frac{P(x)}{Q(x)}$. JS is always finite, bounded between 0 (identical distributions) and $\log 2$ (base-$e$), and admits a metric square root (Lin, 2002; Endres & Schindelin, 2003). We report values in $[0, 1]$ by dividing by $\log 2$.

### 5.2 Benchmark Experimental Settings

#### 5.2.1 Missingness Mechanism and Ratio

The effectiveness of missing data imputation methods is strongly influenced by factors such as the missingness mechanism and ratio. To rigorously evaluate imputation methods, we introduce missing values under 3 missingness mechanisms: MCAR, MAR, and MNAR. The missingness implementation details are provided in Section 4.3. We create versions of the dataset with missingness ratios of 10%, 20%, 30%, 40%, and 50%. The missingness ratio is calculated as the fraction of all entries that are masked, and each feature gets roughly the same fraction of its values missing, though in MAR/MNAR this can vary slightly due to the conditioning. Each missing scenario is generated with five samples and then fixed, so all methods are evaluated on the exact same missing data patterns for fairness.

#### 5.2.2 Imputation Methods

We provide a comprehensive benchmark of 14 widely used imputation methods across four categories: (1) a statistical baseline — Mean/Mode imputation; (2) distribution-matching methods such as **MOT** (Muzellec et al., 2020), which uses optimal transport to align observed and imputed distributions; (3) iterative machine learning methods including **MICE** (Van Buuren & Groothuis-Oudshoorn, 2011), **MIRACLE** (Kyono et al., 2021), **SoftImpute** (Hastie et al., 2015), and **MissForest** (Stekhoven & Bühlmann, 2012); and (4) deep generative models — **MIWAE** (VAE) (Mattei & Frellsen, 2019), **GAIN** (GAN) (Yoon et al., 2018), **DSAN** (self-attention) (Lee & Kim, 2023), and **TabCSDI** (diffusion) (Zheng & Charoenphakdee, 2022). We also include three recent state-of-the-art approaches: **ReMasker** (Du et al., 2024), **HyperImputer** (Jarrett et al., 2022), and **DiffPuter** (Zhang et al., 2025), along with a DSAN variant (**DSN**) without attention to assess its contribution. Implementation details and hyperparameters are in Appendix A.8.

#### 5.2.3 Imputation Performance

For each dataset with missingness, 80% of samples are used for training and 20% for testing. All methods are trained on the training set and then used to impute both in-sample and out-of-sample data. Imputation performance is measured using **RMSE** for continuous and **F1 score** for categorical variables to provide a balanced evaluation given class imbalance. The RMSE is computed on standardized inputs (zero mean, unit variance) based on training-set statistics, and **accuracy** for categorical variables is also reported as a supplementary metric.

#### 5.2.4 Downstream Task Performance

To robustly assess the downstream impact of imputation, we test two task types: **classification** and **regression**. For classification, all three datasets are evaluated with Logistic Regression, Random Forest (RF), XGBoost and LightGBM models; for regression, only SubSDIC is used, with Linear Regression, Random Forest (RF), XGBoost and LightGBM models. Using multiple models reduces model-specific bias. Models are trained on complete training data and evaluated on imputed test sets. For classification, we report the **ROC-AUC degradation**—the drop in ROC-AUC from the fully observed test set—as the main metric, while **accuracy** is provided in the supplement. For regression, we report the **RMSE increase**, the percentage rise in RMSE relative to the fully

observed test set. Smaller ROC-AUC degradation or RMSE increase indicates better imputation quality and stronger preservation of predictive signal.

### 5.2.5 Runtime and Efficiency

To provide practical insight for real-world deployment under resource constraints, we report the total wall-clock time for each method, including both training and imputation. For iterative algorithms such as MICE and MissForest, time covers all iterations; for deep learning models, it includes all training epochs. Appendix A.3 details the experimental setup. Each experiment is repeated five times, and we report mean values and standard deviations as final metrics.

## 5.3 Ranking Consistency

### 5.3.1 Across Datasets

We assess cross-dataset performance consistency using Kendall's coefficient of concordance $W$ as described by Abdi (2007), computed over the 13 methods common to all datasets. Although the full benchmark includes 14 methods, DiffPuter could not be executed on the proprietary CPHS dataset hosted on a CPU-only server and is therefore excluded from the consistency analysis. For $k$ datasets and $N$ methods, let $R_{ij}$ be the rank of method $i$ on dataset $j$, $R_i = \sum_{j=1}^{k} R_{ij}$ the aggregate rank, and $\overline{R} = \frac{k(N+1)}{2}$. Then

$$W = \frac{13}{k^2(N^3 - N)} \sum_{i=1}^{N} (R_i - \overline{R})^2. \tag{4}$$

$W \in [0, 1]$ (0 = no agreement; 1 = perfect concordance). For $k > 2$, $k(N-1)W$ is asymptotically $\chi^2_{N-1}$ under independence (Abdi, 2007). In our study, $k = 3$ datasets and $N = 13$ methods. We use common thresholds: strong ($W > 0.70$), moderate ($0.50 \leq W \leq 0.70$), and weak ($W < 0.50$) agreement (De Maere et al., 2022).

### 5.3.2 Across Downstream Task Models

To quantify the ranking consistency of imputation methods' rankings across different downstream models, we employ Kendall's coefficient of concordance $W$, defined in Eq. 4. This metric enables us to assess whether the relative ordering of imputation methods remains stable across downstream tasks and model families, independent of the absolute performance values.

## 6 Results and Analysis

This section begins with distribution comparison results. We then present a multi-metric benchmark of 14 imputation methods, covering imputation performance, downstream task performance, and computational efficiency. In all figures, the numbers after method names in the legend indicate the average rank of that metric across missingness ratios (shown for the top nine methods only). Full hyperparameter settings and complete results for all datasets and metrics are provided in Appendix A.8 and in the supplementary material. Finally, we quantify the performance consistency between datasets using the Kendall coefficient of concordance computed on the 13 methods available on all datasets.

## 6.1 Distribution Comparison Results

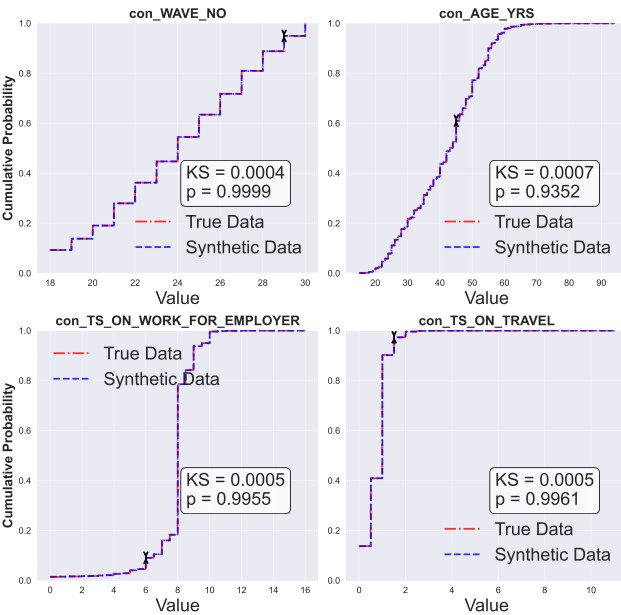

Figure 2: Empirical CDF comparison between CPHS (red) and SynthCPHS (blue) for four representative continuous variables. The double-headed arrow indicates the KS gap $D_{n,m}$.

- **Continuous variables (KS).** For each variable we compute the two-sample KS statistic $D_{n,m}$ in Eq.(2), comparing CPHS and SynthCPHS. The *KS* values displayed in Fig. 2 are exactly this maximum CDF gap $D_{n,m}$. In most cases $D_{n,m} < 10^{-3}$ and the corresponding two-sided $p$-values are close to 1, providing no evidence against the null of identical distributions.

- **Categorical variables (JS).** For each categorical variable we compute the normalized JS divergence, defined as $\mathrm{JS}_{[0,1]} = \frac{\mathrm{JS}(\hat{P}\|\hat{Q})}{\log 2}$, where $\mathrm{JS}(\hat{P}\|\hat{Q})$ is defined in Eq.(3) and we simply rescale it to $[0,1]$. Across all examined variables, $\mathrm{JS}_{[0,1]} = 0$ (to numerical precision), indicating identical empirical distributions between CPHS and SynthCPHS; therefore, we omit plots.

Detailed per-variable values for both continuous and categorical features are provided in the supplementary material.

## 6.2 Imputation Performance

Figures 3 and 4 compare in-sample and out-of-sample performance across methods for continuous (RMSE) and categorical (F1 score) variables, respectively. We observe four key findings:

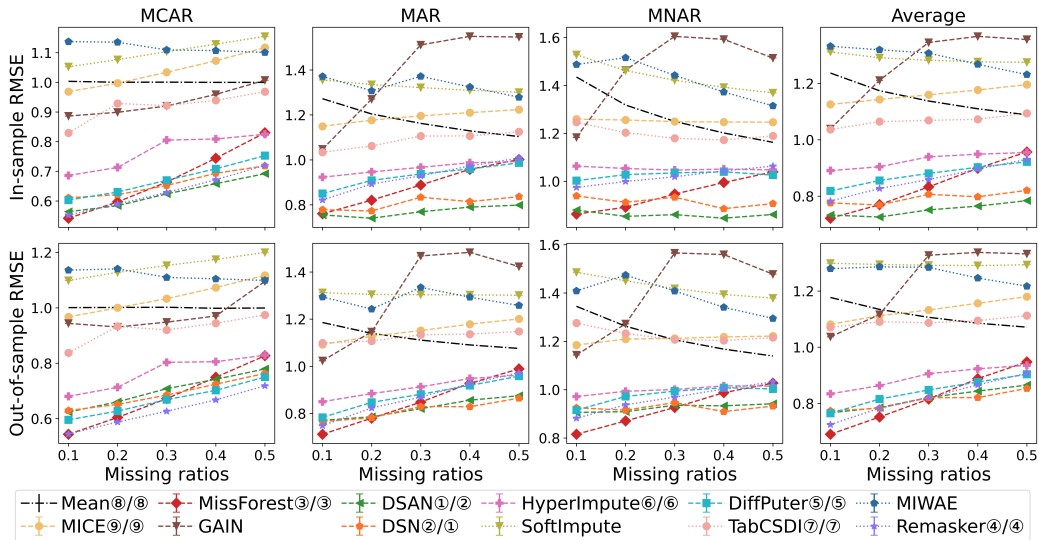

Figure 3: Continuous variable imputation performance. Top row: in-sample; bottom row: out-of-sample. Lower RMSE is better. MIRACLE and MOT are omitted due to excessively high RMSE. Legend entries show in-sample/out-of-sample ranks (first/second number).

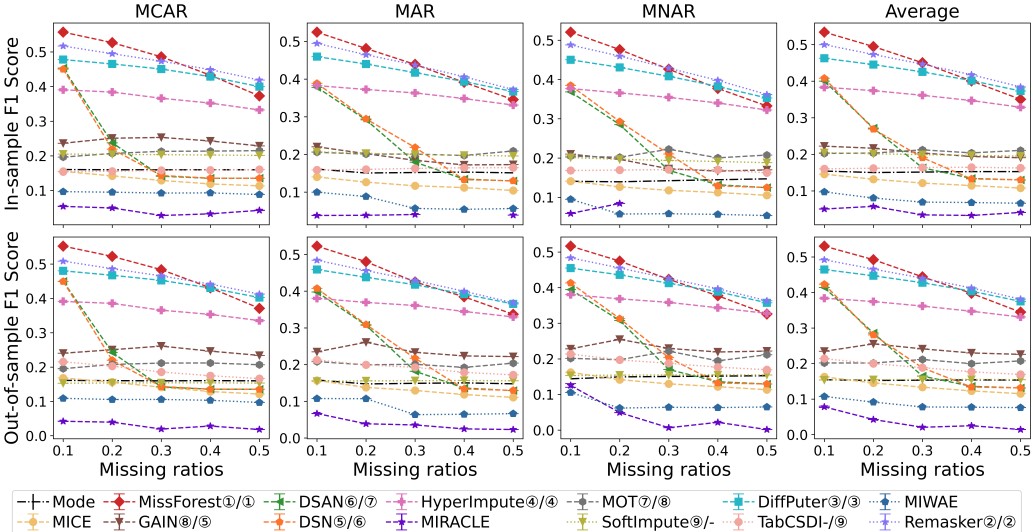

Figure 4: Categorical variable imputation performance. Top row: in-sample; bottom row: out-of-sample. Higher F1 is better. MIRACLE in-sample points are missing at some missingness ratios due to failure. Legend entries show in-sample/out-of-sample ranks (first/second number).

- **Missingness Mechanism Impact.** MCAR yields lower error (lower RMSE and higher F1) than MAR or MNAR, reflecting the challenge of imputing structured missingness. For MAR and MNAR, **RMSE** does not always increase with higher missingness ratios as one might typically expect.

  Specifically, for methods like Mean, SoftImpute, and MIWAE, RMSE decreases as missingness increases. As detailed in **Appendix A.6**, we empirically verified that this is due to the dilution of extreme values inherent in the logistic missingness generation mechanism (Algorithm 1): at low missingness ratios (e.g., 10%), the mechanism selectively masks outliers (which are highly erroneous to impute with the mean); however, at high ratios (e.g., 50%), the missing set expands to include many data points near the global mean, thereby reducing the average RMSE.

  In contrast, **F1 scores** for categorical data typically exhibit two trends: either remaining nearly constant, indicating a failure to learn patterns, or decreasing linearly. An exception is observed in DSAN and its variant DSN, where F1 scores drop sharply under high missingness, suggesting instability in such settings.

- **Method Comparison.** For **continuous variables** (Figure 3), MissForest typically performs best at low missingness ratios (e.g., 10%) but its performance steadily declines as the missingness ratio increases. In contrast, deep learning methods such as ReMasker, DSAN, and DSN outperform MissForest under MCAR, MAR, and MNAR when the missingness ratio is high. For **categorical variables** (Figure 4), MissForest continues to outperform other models at low missingness ratios regardless of the missingness mechanism. However, as the missingness increases, ReMasker and DiffPuter begin to achieve the highest F1 scores. DSAN and DSN, while strong performers for continuous variables, show noticeably worse performance on categorical data, with F1 scores degrading significantly as missingness increases. In summary, MissForest is highly sensitive to the missingness ratio rather than to the type of missingness or variable; it is among the top performers under low missingness (up to 20%), but deep learning methods often become more effective as missingness becomes more severe.

- **Self-Attention Analysis.** As shown in Figures 3 and 4, DSN outperforms DSAN in average ranking across most cases, suggesting limited benefit from the self-attention mechanism. In particular, as shown in Figure 3, while DSAN achieves the best in-sample performance for continuous imputation, DSN outperforms it on average in out-of-sample evaluations, indicating that the attention layer in DSAN introduces a higher risk of overfitting compared to DSN, a phenomenon also observed in a previous study by Dehimi & Tolba (2024).

- **Overfitting Assessment.** Out-of-sample RMSE and F1 scores closely match in-sample results, indicating minimal overfitting for most methods.

**Failure Analysis: MIRACLE and MOT.** A notable observation in our benchmark is the instability and high error rates of MIRACLE and MOT. We attribute these failures to three key factors:

- **MOT: Sinkhorn Instability and Geometric Mismatch.** MOT relies on Sinkhorn iterations to approximate Wasserstein distances. This approach is known to suffer from

numerical instability if the entropy regularization parameter ($\epsilon$) is not meticulously tuned for each dataset (Feydy et al., 2019). Furthermore, MOT relies on Euclidean distances in the data space, which Zhao et al. (2023) argue fails to capture the complex manifold structure of tabular data, resulting in geometrically close but semantically wrong imputations.

- **MIRACLE: Optimization Complexity and Divergence.** MIRACLE involves a multi-objective optimization problem involving causal DAG learning and moment matching regularizers. First, this introduces high hyperparameter sensitivity; as shown in Zhao et al. (2023) (e.g., their Appendix Figures 8-9), MIRACLE frequently yields "empty results" (execution failure) or extreme errors when default hyperparameters do not match the data scale. Second, MIRACLE relies on iterative refinement from a baseline (e.g., Mean imputation). In our datasets, initial baselines often have large errors (see Appendix A.6); this poor initialization can lead the causal graph learning astray, causing the refinement loop to diverge even under MAR mechanisms.

- **MIRACLE: Assumption Violation in MNAR.** Finally, specifically for MNAR, MIRACLE assumes the absence of self-masking missingness (Assumption 3 in Kyono et al. (2021)). Our MNAR settings inherently involve self-masking. Violating this assumption renders the moment regularizer invalid, further destabilizing the model.

## 6.3 Downstream Task Performance

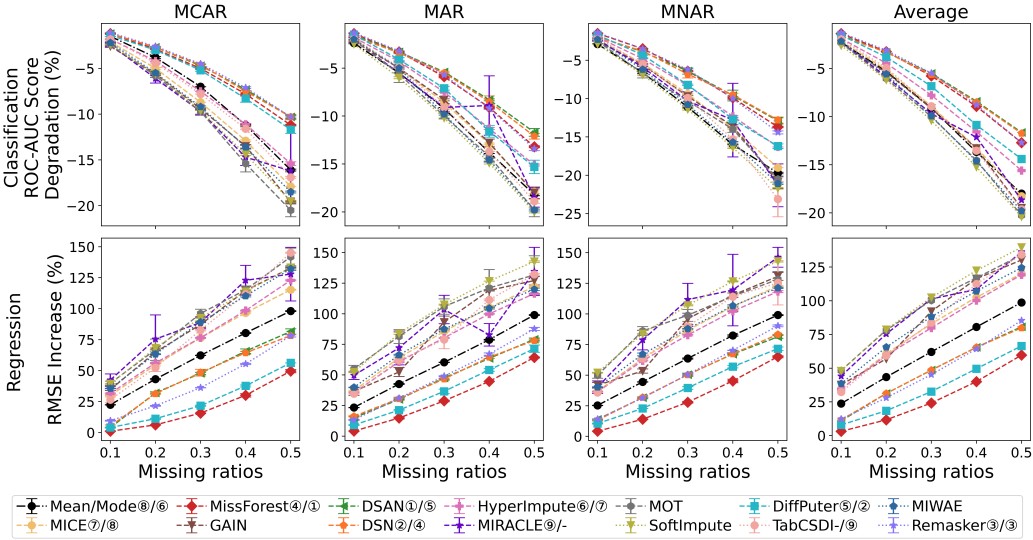

Figure 5: Top: ROC-AUC degradation for downstream classification using Random Forest. Bottom: RMSE increase for downstream regression using Random Forest. Lower ROC-AUC degradation and smaller RMSE increase indicate better preservation of predictive performance. Legend entries show classification performance/regression performance ranks (first/second number).

As shown in Figure 5, we report the downstream impact of imputation on both classification and regression tasks, using Random Forest models as illustrative examples. Unless otherwise specified,

downstream models are run with library default hyperparameters, with only minimal necessary adjustments (e.g., setting `class_weight="balanced"` for classification) and a fixed random seed of 42. All explicitly chosen hyperparameters are documented in the released scripts in our code repository; any parameter not listed there is kept at its default value.

- **Missingness Mechanism.** The results show that missingness under MCAR and MAR leads to slightly lower performance degradation than MNAR. ROC-AUC scores decrease and RMSE increases with higher missingness ratios, highlighting the adverse effect of missing data on downstream performance.

- **Robustness Analysis.** A clear alignment is observed between raw imputation quality (RMSE/F1) and downstream results: methods with lower degradation also achieve top imputation scores. For instance, DSAN and DSN perform best on continuous variables, while ReMasker, MissForest, and DiffPuter excel in categorical imputation, all showing minimal downstream impact. Those top-performing imputation methods maintain their rankings in both regression and classification settings. In two other datasets, while some methods' rankings vary, ReMasker and HyperImpute consistently perform well across all datasets.

- **Sensitivity Analysis.** Figure 5 reports the downstream performance using Random Forest models; similar trends are observed when we replace Random Forest with Linear Regression, XGBoost, or LightGBM (see the "Experiment_result" folder in the supplementary material). To more directly assess the stability of method rankings across different downstream models, we visualize the ranks of all 14 imputers using bump charts for regression and classification in Figure 6a and Figure 6b, respectively. Across Logistic/Linear Regression, Random Forest, XGBoost, and LightGBM, the curves in both bump charts are largely parallel, with only minor local crossings. Top-performing methods remain in the upper part of the ranking for all four models, while the weakest methods stay in the lower part, and no imputer exhibits dramatic shifts from top to bottom or vice versa.

  We further quantify this consistency using Kendall's $W$ defined in Eq. 4. For the classification task, the rankings induced by 4 models yield a Kendall's $W = 0.91$; for the regression task, the rankings give an even higher Kendall's $W = 0.95$. These values indicate strong concordance among the four downstream models in both settings. Taken together, the bump charts and Kendall's $W$ statistics suggest that our main conclusions about the relative performance of imputation methods are not sensitive to the particular choice of downstream model or task type.

### 6.4 Computational Efficiency

As shown in Figure 7, the runtime efficiency analysis indicates that the imputation time is relatively stable across varying missingness ratios and missing mechanisms for most methods. Statistical methods (e.g., Mean/Mode, MICE) and traditional machine learning methods (e.g., HyperImpute, MissForest) demonstrate significantly faster performance, approximately an order of magnitude faster than deep learning-based methods. Among these methods, MissForest stands out for its strong performance on continuous variables at low missingness ratios, solid categorical imputation accuracy, and exceptional time efficiency, making it well suited for large-scale practical applications.

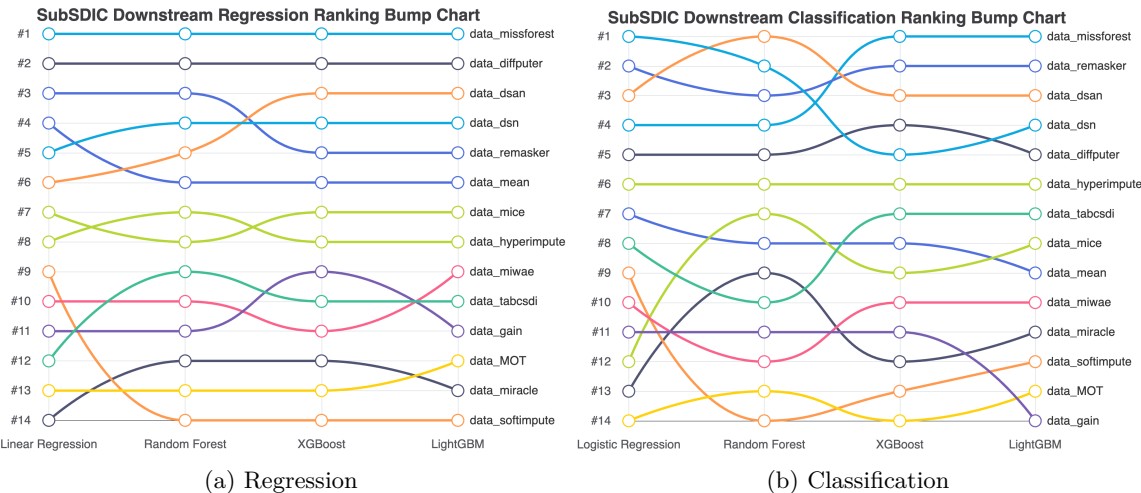

(a) Regression  (b) Classification

Figure 6: SubSDIC Downstream Task Ranking Bump Chart. For regression, Kendall's $W = 0.9473$ across 4 models; and for classification, Kendall's $W = 0.9132$ across 4 models. The overall ordering is highly consistent across downstream models.

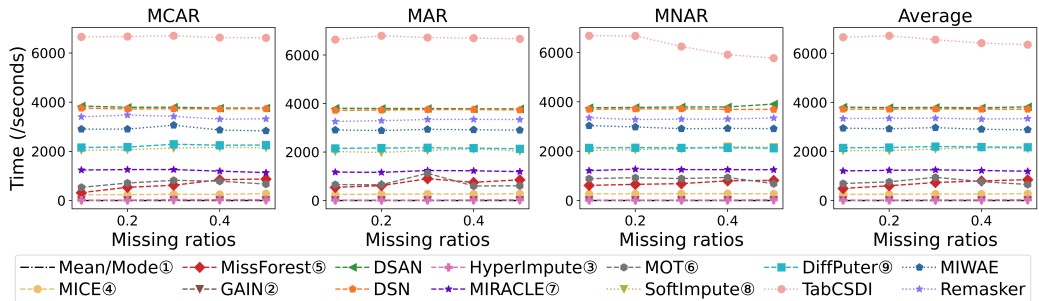

Figure 7: Comparison of imputation runtime.

For more complex missingness scenarios, ReMasker proves to be a powerful imputation method for both continuous and categorical data, offering competitive efficiency compared to other deep learning approaches and resulting in minimal degradation in downstream task performance.

## 6.5 Consistency Analysis

We evaluate the consistency of the rankings across the three datasets using Kendall's coefficient of concordance ($W$) described in Eq.(4). As shown in Table 2, agreement is strong in scenarios A and D, moderate in B, and lower in C. Considering the generally consistent performance across scenarios, we center our discussion above on **SubSDIC**. This dataset is publicly accessible, facilitates both classification and regression tasks, represents the other datasets well, and prevents repetitive reporting across all three.

Table 2: Ranking agreement across 3 datasets measured by Kendall's $W$ (higher is better).

| Scenario | $W$ | Consensus |
|---|---|---|
| A: RMSE (numeric, out-of-sample) | 0.857 | Strong |
| B: F1 (categorical, out-of-sample) | 0.629 | Moderate |
| C: ROC-AUC degradation (RF) | 0.411 | Weak |
| D: Time efficiency | 0.904 | Strong |

## 6.6 Analysis

Overall, by aggregating the average rankings of 14 imputation methods (13 on the CPHS dataset) on three missingness mechanisms and five missingness ratios, across three datasets and four representative tasks (1) Imputation RMSE rank, (2) Imputation F1 rank, (3) downstream regression rank, and (4) downstream classification rank using Random Forest, we find that HyperImpute and MissForest consistently achieve the best overall performance with a clear margin over all other methods. This suggests that although deep learning–based models such as DSAN, DSN, and ReMasker perform competitively, traditional methods like HyperImpute and MissForest deliver more consistent and superior overall results. Our findings therefore reinforce the strong practical competitiveness of traditional machine learning–based methods for missing data imputation, consistent with prior studies (Lalande & Doya, 2022; Zhang et al., 2025; Suh & Song, 2023; Jolicoeur-Martineau et al., 2024; Jäger et al., 2021). Furthermore, our benchmark also suggests that incorporating attention layers may increase the risk of overfitting, in line with the observations of Dehimi & Tolba (2024).

## 6.7 Practical Recommendations

Based on our benchmarking results, we offer the following guidelines for method selection:

**Optimal Balance of Efficiency and Performance:** For general applications, **MissForest** offers the best trade-off between accuracy and efficiency. It achieves superior raw imputation performance at low missingness ratios and demonstrates exceptional stability in downstream tasks. While HyperImpute is also efficient and effective at low ratios, its downstream utility is less consistent compared to MissForest.

**Prioritizing Imputation Performance:** When prioritizing performance over computational cost, deep learning methods outperform baselines as missingness increases (typically $\geq 30\%$):

- Categorical Variables: At high ratios ($\geq 40\%$), **ReMasker** achieves the highest F1 scores, followed by **DiffPuter**.

- Continuous Variables: Under MCAR, **ReMasker** remains the top performer. However, for complex mechanisms (MAR/MNAR) at high ratios, **DSAN** and **DSN** provide the most accurate reconstruction.

In summary, we suggest a hierarchical strategy: use **MissForest** for low missingness ($< 30\%$) due to its efficiency and robustness; switch to deep learning beyond 30% when imputation performance is preferred. Specifically, **ReMasker** and **DiffPuter** are robust for general/categorical data, while **DSAN** and **DSN** excel in continuous imputation under complex patterns. Regarding downstream

predictive stability, **MissForest, ReMasker, DiffPuter, DSAN, and DSN** constitute the top-5 methods that most reliably preserve predictive signals. To ensure the robustness of these recommendations, we conducted rigorous statistical significance tests across varying scenarios; detailed analysis and p-values are provided in Appendix A.9.1.

## 7 Conclusion

**Conclusion:** This work presents a comprehensive benchmark study across three large-scale socioeconomic survey datasets—both real and synthetic—that reflect key characteristics of the domain: longitudinal, hierarchical, large-scale, and non-i.i.d. Using these datasets, we systematically evaluate 14 diverse imputation methods under controlled missingness mechanisms, varying missingness ratios, and across continuous and categorical variables. Beyond imputation accuracy and downstream task performance, we also assess computational efficiency, providing a well-rounded evaluation of each method's practicality.

Our results confirm the strong performance of classical approaches observed in prior studies, while emphasizing the value of multimetric evaluation, including downstream task impact and efficiency, for understanding real-world applicability. The proposed benchmark offers a realistic, robust testbed for missing data research in structured socioeconomic contexts. By releasing the SubSDIC dataset and evaluation framework, we support reproducible research and foster progress in addressing complex missingness patterns in the survey domain.

**Limitations & Future Work:** While CPHS and SynthCPHS provide robust validation of our conclusions, third-party licensing restrictions prevent public dataset release. We note that certain baselines were evaluated using default hyperparameters, though we acknowledge this may conservatively estimate their potential. Graph-based imputers such as GRAPE and IGRM are excluded because their public implementations perform imputation in a learned graph-embedding space rather than the original feature space. Predictions are returned as node and edge representations, and reconstruction losses (e.g., RMSE/MAE) are computed in the latent space. Our benchmark requires per-variable imputations in the original domain and trains downstream models on the reconstructed features. Making these methods compatible would require adding an extra layer that maps graph embeddings back to each column (including discrete variables), thereby violating our fixed-budget and fairness constraints. Finally, we recognize the substantial computational demands of comprehensive benchmarking, particularly for modern deep architectures. Future work will (1) release this framework as an open-source Python package and establish a community leaderboard on PapersWithCode for standardized evaluation, (2) develop prediction models based on heterogeneous graphs that might be better aligned with the structural properties of socioeconomic survey data, and compare graph-based imputers within that study, and (3) continuously incorporate emerging imputation techniques.

### Acknowledgments

The authors would like to acknowledge the use of the University of Oxford Advanced Research Computing (ARC) facility in carrying out this work (DOI: 10.5281/zenodo.22558). This work was also supported by the German Federal Ministry of Research, Technology and Space (BMFTR) under project Eventful (grant number 01IW23005). SF was supported by the EPSRC (grant

EP/V002910/2). AC acknowledges funding from a UKRI AI World Leading Researcher Fellowship (grant EP/W002949/1).

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

# A   Appendix

**Appendix Contents**

## A.1   Ethical Considerations and Limitations

SynthCPHS was generated with the goal of maintaining data utility while ensuring strong privacy protection. We leveraged the `synthpop` framework for fully synthetic data generation, an approach known for its statistical disclosure control properties (Nowok et al., 2017). By design, the synthesized records contain no actual individuals, and thus the risk of re-identification or sensitive attribute disclosure is extremely low. Prior evaluations support this: Nowok et al. (2017) note that fully synthetic data pose minimal disclosure risk, and an independent risk analysis by Elliot (2015) similarly found the disclosure risk in a synthpop-generated dataset to be "very small". In practice, additional safeguards (such as excluding any accidentally replicated unique cases) can be applied to further reduce even the perceived risk of identification.

Nevertheless, this synthetic dataset is not intended as a substitute for the original CMIE CPHS data. For real-world policy and socioeconomic research, direct access to the authentic CPHS data remains indispensable, as only the original data can provide fully reliable and legally accountable insights for decision-making.

### A.2 Code

Code is available here: GitHub

**Reproducibility Guide:** To facilitate reproduction, we provide the **SubSDIC** dataset in the supplementary material. Researchers can reproduce the full benchmark results on SubSDIC by following the README file included in the repository. The workflow requires only minimal configuration (updating file paths marked in the scripts) to execute preprocessing, imputation, and evaluation.

**Data Availability Note:** Please note that reproducibility is restricted to the SubSDIC dataset. The CPHS dataset is proprietary, and the SynthCPHS dataset, due to its high statistical fidelity to the original commercial data, is also subject to distribution restrictions. Consequently, while the codebase supports all datasets, external execution is currently limited to SubSDIC.

### A.3 Configurations

We conduct all experiments with the following setup:

- **Operating System:** Rocky Linux 8 (a rebuild of Red Hat Enterprise Linux 8)

- **CPU:** 2x AMD EPYC 7763 64-Core Processor 1.8GHz (128 cores in total)

- **RAM:** 1000 GiB

- **GPU:** 4x NVIDIA A100-SXM-80GB GPUs (each with 6912 FP32 CUDA cores)

- **Interconnect:** Dual-rail Mellanox HDR200 InfiniBand

- **Cluster:** 90 Dell PowerEdge XE8545 servers

- **Software:** CUDA 11.4, Python 3.9.20, PyTorch Paszke (2019) 2.6.0

### A.4 Datasets

We include only those datasets that have been used in at least three imputation studies. The full names and links for these datasets are provided in Table 3.

### A.5 SubSDIC: Feature Descriptions and Construction Details

Table 4 lists the 19 variables, including the target variables, in the dataset along with the corresponding SDIC "fake" survey questions created to collect the data. Features prefixed with "cat_" indicate categorical variables, while those prefixed with "con_" indicate continuous variables.

Table 3: Related imputation benchmark datasets.

| Dataset (link) | Full name | Brief context |
|---|---|---|
| California Housing | California Housing Prices(Zhang et al., 2025; Du et al., 2024; Jarrett et al., 2022) | Regression of median house value for California districts, using numeric socioeconomic and geographic covariates. |
| Letter Recognition | Letter Recognition (Zhang et al., 2025; Du et al., 2024; Jarrett et al., 2022; Yoon et al., 2018) | Multiclass classification of handwritten letters based on extracted shape / pixel features. |
| Credit Card Clients | Default of Credit Card Clients (Zhang et al., 2025; Du et al., 2024; Yoon et al., 2018) | Binary classification of credit-card default using demographic, repayment and bill-statement variables. |
| Online News | Online News Popularity (Zhang et al., 2025; Yoon et al., 2018) | Regression / classification of article popularity from content and metadata features (social media, keywords, etc.). |
| Concrete Strength | Concrete Compressive Strength (Du et al., 2024; Jarrett et al., 2022; Zheng & Charoenphakdee, 2022) | Regression of concrete compressive strength based on mixture composition and curing age. |
| Wine Quality | Wine Quality (Du et al., 2024; Jarrett et al., 2022; Zheng & Charoenphakdee, 2022) | Quality prediction of red/white wines from physicochemical measurements (regression / ordinal classification). |
| Diabetes Health | Diabetes Health Indicators (Du et al., 2024; Jarrett et al., 2022; Zheng & Charoenphakdee, 2022) | Binary classification of diabetes status using health survey indicators (BMI, lifestyle, comorbidities). |
| SpamBase | SpamBase (Du et al., 2024; Jarrett et al., 2022; Yoon et al., 2018) | Binary classification of email spam based on word and character frequency features. |

### A.6 Analysis of Missing Value Extremeness and RMSE Trends

In Section 6.2, we observed a counterintuitive trend for methods relying on mean-centering, where the RMSE decreases as the missingness ratio increases under MAR and MNAR mechanisms. To validate that this is a structural property of the missingness generation rather than an experimental artifact, we conducted a quantitative analysis of the "extremeness" of the missing values.

### A.6.1 Methodology

We define the **extremeness** of a missing value as its absolute standardized deviation from the global mean (Z-score). For a continuous feature $j$ with mean $\mu_j$ and standard deviation $\sigma_j$, the

Table 4: Detailed Feature Description of SubSDIC

| Feature Name | Question Construct (H=households, I=individuals) |
| --- | --- |
| cat_hid | Household identifier / Machine Generated (H) |
| cat_geo1 | Geographic area - Admin 1 (H) |
| cat_geo2 | Geographic area - Admin 2 (H) |
| cat_urbrur | Urban or rural indicator of household location (H) |
| con_hhsize | Household size, i.e., number of individuals in the household (H) |
| cat_statocc | Does the household own, rent, or occupies this dwelling for free? (H) |
| con_exp_09 | How much does the household spend per year on? (H) |
| con_exp_10 | How much does the household spend per year on? (H) |
| con_tot_exp | Total monthly household expenditure across all categories (H) |
| cat_relation | What is the relationship of [name] to the head of household? (I) |
| cat_sex | Is [name] male or female? (I) |
| con_age | How old is [name]? (I) |
| cat_marstat | What is [name's] marital status? (I) |
| cat_religion | What is the religion of [name]? (I) |
| cat_school_attend | Is [name] attending school or preschool? (I) |
| con_yrs_school | How many years has [name] attended school? (I) |
| cat_act_status | What is [name's] status of activity? (I) |
| cat_occupation | What is/was [name's] main occupation? (I) |
| cat_educ_attain | What is the highest level of school that [name] has completed? (I) |

extremeness score $E$ for the set of missing entries $\mathcal{M}_j$ at a specific missingness ratio is calculated as:

$$E = \frac{1}{|\mathcal{M}_j|} \sum_{i \in \mathcal{M}_j} \frac{|x_{ij} - \mu_j|}{\sigma_j} \tag{5}$$

A higher $E$ indicates that the missing values are, on average, further from the center of the distribution (i.e., outliers). We computed this metric across all continuous features in the SubSDIC dataset under MCAR, MAR, and MNAR mechanisms across five missingness ratios.

### A.6.2 Results and Interpretation

Figure 8 illustrates the relationship between the missingness ratio and the extremeness of the masked values.

The results reveal distinct behaviors:

- **MCAR (Blue Line):** The curve is flat, indicating that the distribution of missing values is consistent with the global distribution ($E \approx 0.75$) regardless of the ratio. This explains why Mean Imputation performance is relatively stable under MCAR.

- **MAR and MNAR (Orange and Green Lines):** There is a significant downward trend. At a low missingness ratio (10%), the logistic generation mechanism is highly

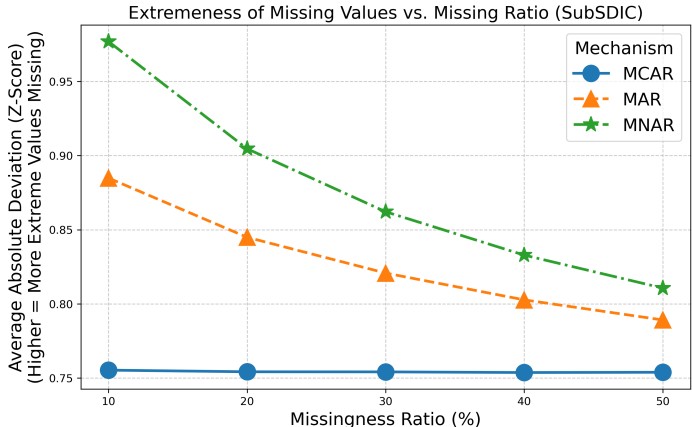

Figure 8: Average absolute deviation (Z-score) of the ground truth values of missing entries across different missingness ratios. Under MAR and MNAR, the "extremeness" of missing data decreases as the ratio increases.

selective, masking primarily observations with extreme values (outliers) that yield the highest probabilities. Consequently, $E$ is high ($> 0.85$). As the ratio increases to 50%, the threshold for missingness lowers, and the mechanism includes a larger proportion of data points lying closer to the global mean, thus lowering the average extremeness ($E$ drops to $\approx 0.80$).

### A.7    Temporal Dependency Analysis

To evaluate whether our benchmark datasets and missingness mechanisms account for the longitudinal characteristics of socioeconomic surveys, we performed a quantitative analysis of temporal dependencies using the Autocorrelation Function (ACF).

#### A.7.1    Methodology: Pseudo-Panel Analysis

Since individual household identifiers were removed from the CPHS and SynthCPHS datasets to ensure they are not used as semantic predictive features, we constructed a *pseudo-panel* to analyze temporal signals. Households were aggregated by their Homogeneous Region (HR) (`cat_HR`), a geographical unit designed to cluster households with similar socio-economic characteristics, and by survey round (`con_WAVE_NO`).

To quantify the global temporal dependency, we adopted a two-step approach:

1. **HR-Level Autocorrelation:** First, we computed the Lag-1 autocorrelation ($\rho_k$) for each specific Homogeneous Region $k$ individually. Let $x_{k,t}$ represent the mean value of a feature within region $k$ at wave $t$. The autocorrelation for region $k$ is calculated as:

$$\rho_k = \frac{\sum_{t=2}^{T}(x_{k,t} - \bar{x}_k)(x_{k,t-1} - \bar{x}_k)}{\sqrt{\sum_{t=2}^{T}(x_{k,t} - \bar{x}_k)^2}\sqrt{\sum_{t=2}^{T}(x_{k,t-1} - \bar{x}_k)^2}} \tag{6}$$

where $\bar{x}_k$ is the temporal mean of the feature for region $k$.

2. **Global Aggregation:** We then reported the **average autocorrelation** across all valid regions to represent the dataset-level temporal dependency:

$$\bar{\rho} = \frac{1}{K}\sum_{k=1}^{K}\rho_k \tag{7}$$

where $K$ is the total number of HRs and $K = 102$.

### A.7.2 Baseline Temporal Dependency

We utilized the effect size benchmarks established by Cohen (1988) (Chapter 3), where a correlation of $|r| = 0.30$ is categorized as a **medium** effect and $|r| = 0.50$ as a **large** effect.

As detailed in Table 5, we select the top 10 features ranked by $\bar{\rho}_{\mathrm{orig}}$. The original CPHS data exhibit significant temporal dependencies: income-related features typically show autocorrelation coefficients above 0.30, with some member-level income variables reaching as high as 0.476. These results confirm that the datasets contain moderate-to-strong longitudinal signals, distinguishing them from traditional independent and identically distributed (i.i.d.) tabular benchmarks.

### A.7.3 Impact of Missingness and Imputation

We further analyzed the autocorrelation of the masked (incomplete) data, presenting results at a 30% missingness ratio to assess structural preservation. We define the Relative Drop ($\delta$) using the following equation:

$$\delta = \frac{\bar{\rho}_{\mathrm{orig}} - \bar{\rho}_{\mathrm{mask}}}{\bar{\rho}_{\mathrm{orig}}} \tag{8}$$

where $\bar{\rho}_{\mathrm{orig}}$ and $\bar{\rho}_{\mathrm{mask}}$ denote the average cohort-level Lag-1 autocorrelation in the original and masked datasets, respectively.

Our analysis revealed that under **MAR** and **MNAR**, temporal dependencies are highly preserved, with relative drops often smaller than 5%. In some MNAR scenarios, a slight increase in ACF (negative $\delta$) was observed, likely due to "survivor bias," where the missingness mechanism selectively censors highly volatile observations, artificially smoothing the observed cohort means.

Furthermore, we evaluated the ACF of the datasets with imputation. We observed that most imputation methods yielded ACF values higher than those in the masked data, demonstrating their capability to restore temporal dependencies. Comprehensive results—including ACF values for every feature across the complete dataset, the masked datasets, and the datasets imputed by all 14 methods—are provided in the `Supplementary_Material`.

### A.8 Implementations and Hyperparameters

### A.8.1 Implementation of Models

We implemented all 14 imputation methods based on open access GitHub repository following:

- Mean/Mode: Implemented using the NumPy package.

- MOT (Muzellec et al., 2020): `https://github.com/BorisMuzellec/MissingDataOT`.

Table 5: Comparison of HR-Level Autocorrelation (Lag-1) Before and After Missingness Generation (Missingness Ratio = 30%). The $\delta$ column indicates the percentage decrease in autocorrelation relative to the original data.

| Feature Name | Original | MCAR (30%) | | | MAR (30%) | | | MNAR (30%) | | |
|---|---|---|---|---|---|---|---|---|---|---|
| | ACF ($\bar{\rho}_{\mathrm{orig}}$) | $\bar{\rho}_{\mathrm{mask}}$ | $\delta$ % | Preserved? | $\bar{\rho}_{\mathrm{mask}}$ | $\delta$ % | Preserved? | $\bar{\rho}_{\mathrm{mask}}$ | $\delta$ % | Preserved? |
| con_INC_OF_MEM_FRM_ALL_SRCS_2 | 0.476 | 0.452 | 5.0% | Yes | 0.476 | **0.0%** | **Yes** | 0.483 | **-1.5%** | **Yes** |
| con_INC_OF_MEM_FRM_ALL_SRCS_3 | 0.466 | 0.430 | 7.7% | Yes | 0.451 | 3.2% | Yes | 0.460 | **1.3%** | **Yes** |
| con_INC_OF_MEM_FRM_ALL_SRCS_1 | 0.400 | 0.376 | 6.0% | Yes | 0.378 | 5.5% | Yes | 0.378 | 5.5% | Yes |
| con_TOT_INC_2 | 0.386 | 0.355 | 8.0% | Yes | 0.361 | 6.5% | Yes | 0.372 | 3.6% | Yes |
| con_INC_OF_MEM_FRM_ALL_SRCS_4 | 0.361 | 0.329 | 8.9% | Yes | 0.347 | 3.9% | Yes | 0.346 | 4.2% | Yes |
| con_TOT_INC_3 | 0.355 | 0.335 | 5.6% | Yes | 0.331 | 6.8% | Yes | 0.335 | 5.6% | Yes |
| con_INC_OF_HH_FRM_ALL_SRCS_2 | 0.335 | 0.298 | 11.0% | Partial | 0.312 | 6.9% | Yes | 0.288 | 14.0% | Partial |
| con_INC_OF_HH_FRM_ALL_SRCS_3 | 0.323 | 0.282 | 12.7% | Partial | 0.283 | 12.4% | Partial | 0.278 | 13.9% | Partial |
| con_TOT_INC_1 | 0.297 | 0.282 | 5.1% | Yes | 0.258 | 13.1% | Partial | 0.264 | 11.1% | Partial |
| con_TS_ON_TRAVEL | 0.290 | 0.281 | 3.1% | Yes | 0.282 | 2.8% | Yes | 0.274 | 5.5% | Yes |

- MissForest (Stekhoven & Bühlmann, 2012): Implemented using the missforest package.

- DSAN (Lee & Kim, 2023): `https://github.com/uos-dmlab/Structued-Data-Quality-Analysis/tree/master`.

- DSN (Lee & Kim, 2023): Developed from DSAN by removing the attention layer.

- TabCSDI (Zheng & Charoenphakdee, 2022): `https://github.com/pfnet-research/TabCSDI`.

- Remasker (Du et al., 2024): `https://github.com/tydusky/remasker`.

- DiffPuter (Zhang et al., 2025): `https://github.com/hengruizhang98/DiffPuter`.

- For HyperImputer (Jarrett et al., 2022), MICE (Van Buuren & Groothuis-Oudshoorn, 2011), MIRACLE (Kyono et al., 2021), SoftImpute (Hastie et al., 2015), MIWAE (Mattei & Frellsen, 2019), and GAIN (Yoon et al., 2018), we use implementations at: `https://github.com/vanderschaarlab/hyperimpute`.

The codes for all methods are available in the anonymous GitHub repository.

### A.8.2 Hyperparameter Settings of Models

Most of the methods included in our benchmark recommend using a single set of hyperparameters across different datasets. For such methods, we adopt the default hyperparameters provided in their official GitHub repositories and ensure sufficient training epochs or steps to achieve convergence of the training loss. The anonymous GitHub repository provides the implementation with the default hyperparameters applied across all methods.

### A.8.3 Hyperparameter Sensitivity and Convergence Analysis

To assess the impact of default hyperparameters on our benchmark rankings, we conducted a focused analysis on **GAIN**, a generative model known for its sensitivity to hyperparameter settings. We selected the SubSDIC dataset under MNAR (30% missingness) as the testbed, as GAIN exhibited a notable performance drop in this specific setting (RMSE $\approx$ 1.50) as shown in Figure 3.

**1. Convergence Verification.** First, we verified that the default training duration was sufficient. We extended the training from the default 1,000 epochs to 10,000 epochs and monitored the total training loss. As shown in Figure 9, the loss curves for multiple random seeds stabilize well before 1,000 epochs, confirming that the default settings do not suffer from under-fitting.

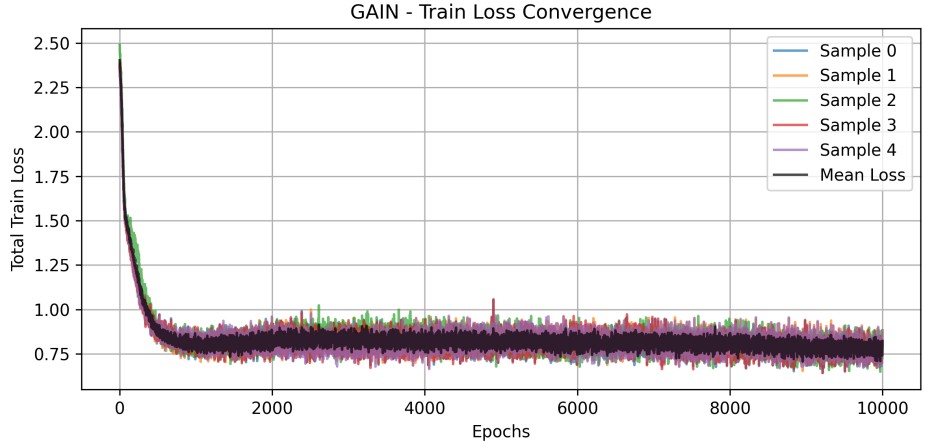

Figure 9: Total training loss convergence of GAIN on SubSDIC (MNAR 30%) over 10,000 epochs. The model reaches convergence well within the default 1,000 epochs.

**2. Grid Search and Ranking Stability.** We performed a grid search over two critical hyperparameters for GAIN:

- `hint_rate`: $\{0.1, 0.3, 0.5, 0.7, 0.9\}$
- `loss_alpha`: $\{20, 40, 60, 80, 100\}$

The results indicated that the optimal configuration (with `hint_rate`=0.9 and `loss_alpha` $\in \{60, 80, 100\}$) achieved an RMSE of **1.20**, a significant improvement over the default RMSE of **1.50**.

**Impact on Benchmarking Conclusions:** While tuning improved GAIN's absolute performance and raised its rank by 3 positions, it remained significantly outperformed by the top-tier methods identified in our benchmark. Specifically, models such as **MissForest, DSAN, DSN, ReMasker, and DiffPuter** all achieved RMSE values **below 1.0** on the same task. This suggests that while hyperparameter tuning is beneficial, the relative performance gap between the most effective tabular imputation methods and others is robust and primarily driven by model architecture capabilities rather than hyperparameter configurations. The experiments results can be found in the folder of the supplementary material named "Experiment_result/SubSDIC/Imputation_performance/Out-sample".

### A.9 Experimental Results

All the experimental results, including KS test, JS divergence, and Kendall's W, can be found in the folder of the supplementary material named "Experiment_result". The following sections present only a selection of representative results which are not listed in the main text.

### A.9.1 Statistical Significance Analysis

To ensure the statistical rigor of our benchmarking results and recommendations, we conducted pairwise t-tests (based on 5 random seeds) for all 14 methods on the **SubSDIC** dataset. This analysis generated 91 pairwise combinations across 15 experimental settings (3 mechanisms $\times$ 5 ratios) and 2 task types (continuous and categorical imputation), yielding a total of 2,730 p-values. We found that **84%** of these comparisons showed statistically significant differences ($p < 0.05$). The full set of p-values is available in the `Supplementary_Material`.

Based on the practical recommendations outlined in Section 6.7, we highlight the statistical verification of the top-performing methods (**MissForest, ReMasker, DiffPuter, DSAN, DSN, HyperImpute**) in four critical scenarios.

**Continuous Variable Imputation (RMSE)** We validated the imputation performance for continuous variables, with detailed p-values provided in Table 6.

1. Low Missingness (10%, MAR/MNAR). **MissForest** is significantly superior to other top-tier models (including ReMasker, DSAN, and HyperImpute) with $p < 0.05$. This statistically confirms its dominance and suitability as a default choice in low-missingness regimes.

2. High Missingness (50%, MAR/MNAR). **DSN** significantly outperforms traditional baselines and other deep models ($p < 0.05$). However, the difference between DSN and **DSAN** is not statistically significant ($p > 0.05$). This aligns with our observation in Figure 3 that their performance is comparable, suggesting that the attention layer in DSAN does not yield a significant advantage over DSN in this specific context.

**Categorical Variable Imputation (F1 Score)** We conducted similar tests for categorical variables (detailed tables omitted for brevity, full results in Supplementary Material):

1. Low-to-Medium Missingness ($\leq 20\%$, All Mechanisms). **MissForest** significantly outperforms all 13 other methods ($p < 0.05$), demonstrating exceptional robustness for categorical data in this regime.

2. High Missingness (50%). **ReMasker** significantly outperforms other methods ($p < 0.05$), with the exception of **DiffPuter**. The difference between ReMasker and DiffPuter is not statistically significant ($p > 0.05$), which is consistent with Figure 4, where both generative models demonstrate similarly high performance at extreme missingness ratios.

### A.9.2 Downstream Task Performance: LightGBM

To further examine the sensitivity of downstream evaluation to model choice, we additionally evaluate LightGBM on the SubSDIC dataset, the only dataset in our benchmark that naturally supports both a classification and a regression task. LightGBM is assessed in two modes:

1. using imputed data produced by each of the 14 imputation methods;

2. using LightGBM's native handling of missing values as a naïve "no-imputation" baseline.

Table 6: P-values for pairwise comparisons of top-tier methods on SubSDIC (Continuous Variables). **Bold** scenarios highlight the superior performance of Method A. ($p < 0.05$ indicates statistical significance).

| Task Type | Scenario | Method A (Hypothesis: Better) | Method B | p-value | Significant? |
|---|---|---|---|---|---|
| *Continuous Variable Imputation (Metric: RMSE)* | | | | | |
| | MAR 10% | MissForest | ReMasker | **0.0418** | Yes |
| | MAR 10% | MissForest | DSAN | **0.0346** | Yes |
| | MAR 10% | MissForest | DSN | **0.0044** | Yes |
| | MAR 10% | MissForest | DiffPuter | **0.0000** | Yes |
| | MAR 10% | MissForest | HyperImpute | **0.0001** | Yes |
| Low Missingness | MNAR 10% | MissForest | ReMasker | **0.0061** | Yes |
| | MNAR 10% | MissForest | DSAN | **0.0222** | Yes |
| | MNAR 10% | MissForest | DSN | **0.0338** | Yes |
| | MNAR 10% | MissForest | DiffPuter | **0.0089** | Yes |
| | MNAR 10% | MissForest | HyperImpute | **0.0015** | Yes |
| | MAR 50% | DSN | ReMasker | **0.0025** | Yes |
| | MAR 50% | DSN | MissForest | **0.0005** | Yes |
| | MAR 50% | DSN | DiffPuter | **0.0078** | Yes |
| | MAR 50% | DSN | HyperImpute | **0.0022** | Yes |
| | MNAR 50% | DSN | ReMasker | **0.0021** | Yes |
| High Missingness | MNAR 50% | DSN | MissForest | **0.0086** | Yes |
| | MNAR 50% | DSN | DiffPuter | **0.0000** | Yes |
| | MNAR 50% | DSN | HyperImpute | **0.0069** | Yes |

As shown in Figure 10, the "no-imputation" (marked as "w/o imputation") baseline yields the **worst** performance across all missingness mechanisms and across most missingness ratios, for both classification and regression tasks. In classification, the ROC-AUC degradation under native missing-value handling is substantially larger than that of any imputed alternative. Likewise, in regression, the RMSE increase under the "w/o imputation" baseline dominates that of all imputed variants.

Notably, even the weakest imputer in our pool outperforms LightGBM's native missing-value strategy under most experimental configuration. This finding provides a clear practical implication: in the context of socioeconomic survey data, explicit imputation is consistently beneficial and should not be replaced by relying solely on LightGBM's built-in handling of missing values.

Together with the sensitivity analyses under linear/logistic regression, Random Forest, and XGBoost in Section 6.3, these results further demonstrate that our conclusions regarding the relative ordering of imputation methods are robust to the choice of downstream predictive model.

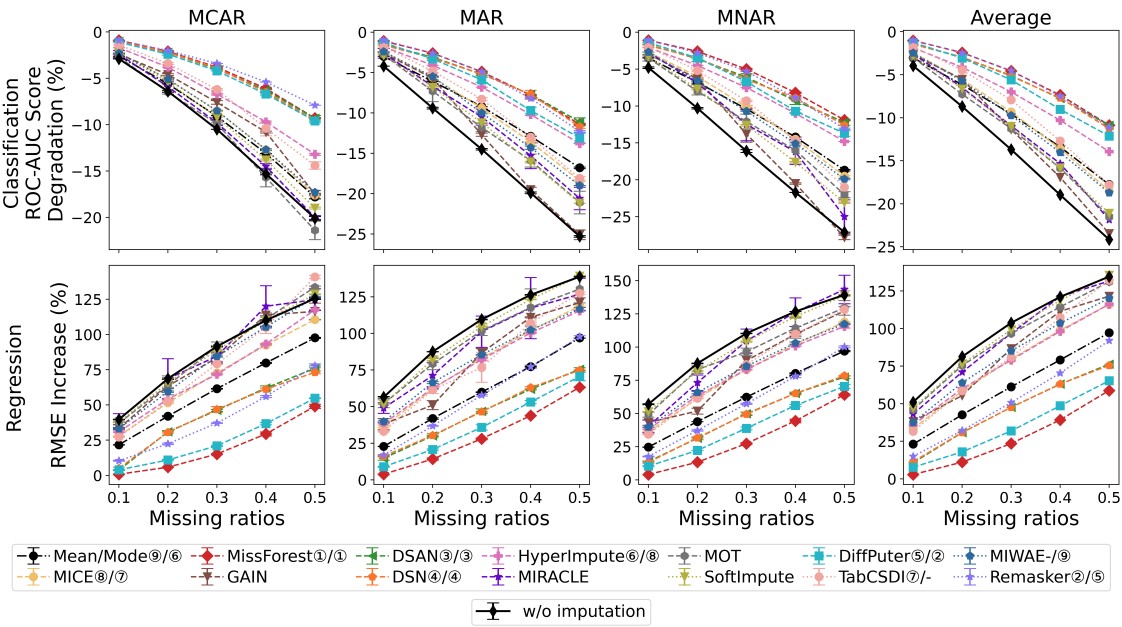

Figure 10: Top: ROC-AUC degradation for downstream classification using LightGBM. Bottom: RMSE increase for downstream regression using LightGBM. Lower ROC-AUC degradation and smaller RMSE increase indicate better preservation of predictive performance. Legend entries show classification performance/regression performance ranks (first/second number).

