# OpenReview forum: "Benchmarking Missing Data Imputation Methods in Socioeconomic Surveys"
_TMLR — Accepted by TMLR_

### Review · Reviewer_Qn8w · 2025-11-18

**Summary Of Contributions:**

This paper presents a large-scale, domain-specific benchmark for missing-data imputation in socioeconomic surveys. The authors construct and evaluate imputation methods on three structured datasets: the proprietary CPHS dataset, a synthetic replica, and a publicly available survey-like dataset. The benchmark systematically compares 14 classical, statistical, and deep-learning imputation algorithms across MCAR/MAR/MNAR mechanisms, five missingness levels, continuous and categorical variables, downstream predictive tasks, and runtime.

Strengths:
- Socioeconomic survey imputation is critical yet under-studied relative to healthcare or UCI-style tabular data. A reproducible benchmark addressing longitudinal, hierarchical, non-IID structures fills a real gap.
- The paper evaluates a large set of methods belonging to different families, across mechanisms, missingness ratios, variable types, and downstream tasks (continuous/categorical imputation + downstream classification/regression).
- Findings reinforce the empirical value of classical methods and highlight when deep learning approaches excel or fail.
- Providing the full evaluation pipeline is a valuable contribution to the community.

Weaknesses:
- The paper repeatedly highlights that socioeconomic surveys are longitudinal, and this is presented as one of the main motivations for creating a domain-specific benchmark. However, the longitudinal structure does not appear to be explicitly modeled, exploited, or stressed in the methodology. Missingness is generated feature-wise via logistic models that do not clearly incorporate time dependencies, nor is it explained whether missingness mechanisms preserve realistic longitudinal patterns. Additionally, none of the evaluated imputation methods are time-aware or sequence-based, and the results section does not analyze whether the temporal dimension influences imputation difficulty. As a result, it is unclear whether the benchmark truly evaluates the challenges unique to longitudinal survey data, or whether the datasets are effectively treated as cross-sectional tables. Clarifying how longitudinal dependencies were handled, both in missingness generation and in model evaluation, would strengthen the contribution and align the experiments more closely with the paper’s stated motivation.
- Some methodological details could improve in clarity, particularly regarding missingness generation and default hyperparameter usage.
- Limited discussion of failure cases (e.g., why MIRACLE or MOT performed so poorly), which would help contextualize results.

**Additional Comments:**

- How sensitive are results to the choice of downstream models? RF/XGB are robust, but some imputation methods that leverage nonlinear interactions may perform differently under linear models.
- Could the benchmark be extended to assess imputation uncertainty? Several deep generative models produce uncertainty estimates and incorporating proper scoring rules could open an interesting dimension.
- Do the authors plan to provide a leaderboard extension of this benchmark?

**Audience:**

Yes

**Audience Explanation:**

The work sits at the intersection of ML for tabular data, imputation, and synthetic data generation. It provides a valuable benchmark for researchers developing new imputation methods, especially those targeting realistic tabular settings.

**Broader Impact Concerns:**

No concern.

**Claims And Evidence:**

Yes

**Claims Explanation:**

The authors use rigorous empirical evaluation and provide extensive metrics. Results are consistent across datasets, and statistical comparisons (KS/JS divergence) support the validity of the synthetic benchmark.

**Requested Changes:**

- Clarify how longitudinal structure is handled. Since longitudinality is highlighted as a main motivation, the paper should explain whether missingness generation preserves time dependencies and whether the evaluation reflects temporal structure.
- Clarify missingness generation mechanisms in the main text. Appendix A.6 provides the generation details, but the paper should explicitly summarize the logistic-model–based procedure for MAR/MNAR, how parameters β are learned, whether the same logistic models are used across datasets or re-estimated each time, how temporary imputations work under MNAR. This is central to the benchmark’s validity and should not rely solely on appendices.
- The paper acknowledges that default hyperparameters were used for many baselines, but this may introduce bias. The authors should clarify which methods were most sensitive to hyperparameters, justify why deep learning models were not tuned, and discuss whether performance rankings might shift with tuning.
- Discuss failure modes of specific models providing short diagnostic section to help contextualize the benchmark.
- The authors use RF and XGBoost, which is reasonable, but ideally they should justify these choices over linear models or neural networks and clarify whether models were tuned or used with defaults. Additionally, comparing against a naïve “no-imputation” baseline or noting widespread gradient boosting methods like LightGBM that natively handle missingness could enrich the evaluation.
- A slightly more detailed explanation of why models like GRAPE or IGRM are incompatible with the evaluation framework would improve clarity.

---

> ### Author Response · Authors · 2025-12-20
> **Response (Part 1)**
>
> We thank the reviewer Qn8w for the constructive feedback. Below, we provide our point-by-point responses.
> * [**C1**]: Clarify how longitudinal structure is handled. Since longitudinality is highlighted as a main motivation, the paper should explain whether missingness generation preserves time dependencies and whether the evaluation reflects temporal structure.
>
> **Response**: We thank the reviewer for this insightful observation. We clarify that while our benchmark focuses on imputation, the CPHS dataset possesses significant longitudinal properties. To address your specific concerns, we conducted a rigorous autocorrelation analysis to validate the existence of temporal dependencies and their preservation under missingness mechanisms.
>
> **1. Validation of Temporal Dependency (Original Data)** Since individual identifiers are removed in the CPHS and SynthCPHS datasets to ensure they are not used as predictive features for downstream tasks (as the ID values carry no information), we performed a pseudo-panel analysis by aggregating households within Homogeneous Regions (HR) in CPHS. We computed the cohort-level lag-1 autocorrelation for key socioeconomic variables.
>
> As shown in Table 5, the original values of the autocorrelation function (ACF), specifically the lag-1 autocorrelation, typically exceed 0.30 for income-related variables, with specific member-level income features approaching 0.50 (e.g., `con_INC_OF_MEM_FRM_ALL_SRCS_2`). It is important to note that the lag-1 ACF is mathematically equivalent to the Pearson product-moment correlation coefficient ($r$) computed between a time series and its lagged version. Therefore, following standard benchmarks in the behavioral sciences$^{[1]}$, where $|r|$ = 0.30 is considered a medium effect size and $|r|$ = 0.50 a large effect size, we confirm that the dataset contains moderate-to-strong longitudinal signals. This distinguishes CPHS from standard i.i.d. tabular benchmarks.
>
> **2. Preservation under Missingness Generation** We further analyzed the autocorrelation of the masked (incomplete) data, reporting results at a 30% missingness ratio to assess structural preservation. We define the **Relative Drop δ** as (ACF$ _ {orig}$ - ACF$ _ {mask}$) \/ ACF$ _ {orig}$.
>
> | Feature Name | Original ACF$ _ {orig}$ | MCAR ACF$ _ {mask}$ | δ (%) | Preserved? | MAR ACF$ _ {mask}$ | δ (%) | Preserved? | MNAR ACF$ _ {mask}$ | δ (%) | Preserved? |
> |-------------|--------------|----------|-------|------------|---------|-------|------------|----------|-------|------------|
> | `con_INC_OF_MEM_FRM_ALL_SRCS_2` | 0.476 | 0.452 | 5.0 | Yes | 0.476 | **0.0** | **Yes** | 0.483 | **-1.5** | **Yes** |
> | `con_INC_OF_MEM_FRM_ALL_SRCS_3` | 0.466 | 0.430 | 7.7 | Yes | 0.451 | 3.2 | Yes | 0.460 | **1.3** | **Yes** |
> | `con_INC_OF_MEM_FRM_ALL_SRCS_1` | 0.400 | 0.376 | 6.0 | Yes | 0.378 | 5.5 | Yes | 0.378 | 5.5 | Yes |
> | `con_TOT_INC_2` | 0.386 | 0.355 | 8.0 | Yes | 0.361 | 6.5 | Yes | 0.372 | 3.6 | Yes |
> | `con_INC_OF_MEM_FRM_ALL_SRCS_4` | 0.361 | 0.329 | 8.9 | Yes | 0.347 | 3.9 | Yes | 0.346 | 4.2 | Yes |
> | `con_TOT_INC_3` | 0.355 | 0.335 | 5.6 | Yes | 0.331 | 6.8 | Yes | 0.335 | 5.6 | Yes |
>
> Due to space limitations, we present only a subset of the results from Table 5 of the revised manuscript in this response. The results demonstrate that:
>
> - Under **MCAR**, we observe a moderate decay in autocorrelation (approximately a 5–10% drop), consistent with random information loss.
> - Crucially, under **MAR** and **MNAR**, the temporal dependencies are highly preserved. In several cases (e.g., `con_INC_OF_MEM_FRM_ALL_SRCS_2`), we observe a slight increase in ACF (i.e., a negative relative drop). This is likely due to a *survivor bias* effect, where the missingness mechanism selectively censors highly volatile observations, thereby artificially smoothing the observed cohort means.
>
> This analysis confirms that our feature-dependent missingness mechanisms do not reduce the data to random noise, but instead preserve the underlying longitudinal structure. Consequently, our benchmark meaningfully challenges imputation models to implicitly learn and exploit these preserved temporal dependencies.
>
> We have added **Appendix A.7** to the revised manuscript to provide a detailed description of this temporal dependency analysis. Furthermore, we evaluated the ACF of the datasets after imputation. We find that most imputation methods yield ACF values higher than those of the masked data, demonstrating their ability to restore temporal dependencies. Comprehensive results on the CPHS dataset—including ACF values for every feature across the complete dataset, the masked datasets, and the datasets imputed by all 14 methods—are provided in the **Supplementary Material**.
>
> ### **Reference**
> - [1] [Cohen, J. (1988). Statistical Power Analysis for the Behavioral Sciences (2nd ed.). Routledge. ](https://doi.org/10.4324/9780203771587)

---

> ### Author Response · Authors · 2025-12-20
> **Response (Part 2)**
>
> * [**C2**] Clarify missingness generation mechanisms in the main text. Appendix A.6 provides the generation details, but the paper should explicitly summarize the logistic-model–based procedure for MAR/MNAR, how parameters β are learned, whether the same logistic models are used across datasets or re-estimated each time, how temporary imputations work under MNAR. This is central to the benchmark’s validity and should not rely solely on appendices.
>
> **Response**: We sincerely thank the reviewer for highlighting the importance of explicitly detailing the missingness generation process in the main text. We agree that transparency regarding these mechanisms is central to the validity of the benchmark.
>
> We have revised **Appendix A.6** and moved its content to **Section 4.3 (Missingness Generation)** in the revised manuscript to provide a comprehensive summary of the logistic-model-based procedures. In addition, we have added **Algorithm 1** to illustrate the step-by-step generation workflow, explicitly highlighting the differences between the MAR and MNAR mechanisms.
>
> Regarding the specific technical questions raised by the reviewer, we have clarified the following points in the revised text:
>
> 1. **Specification of the parameters $\beta $**: In our synthetic generation framework, the coefficient vector $\beta _ j$ represents the weights assigned to features that influence the missingness of variable $j$. These parameters are *not learned from data*; instead, they are explicitly defined to apply **uniform weights** across influencing covariates:
>
>    - **Under MAR:** The missingness of variable $j$ depends on all *observed* variables except itself. Accordingly, the weights are set as  $\beta _ k$ = 1 / ($d$ − 1) for all $k ≠ j$ (where $d$ denotes the total number of features), and $\beta _ j$ = 0.
>
>    - **Under MNAR:** The missingness depends on all variables, including the variable itself. Therefore, the weights are set uniformly as $\beta _ k$ = 1 / $d$ for all k, allowing the variable’s own value to directly influence its missingness probability.
>
> 2. **Independence of generation models**: We confirm that the missingness generation process is **fully independent** for every experimental configuration. The logistic models (and their associated masks) are generated for each dataset (CPHS, SynthCPHS, SubSDIC), each mechanism, each missingness ratio, and each random seed. They are not reused across datasets.
>
> 3. **Temporary mean imputations**: Temporary mean imputation is a procedure within the generation framework designed to handle pre-existing missing values in the input data. Specifically, if the input data contains missing entries, they are temporarily filled with column means to allow the logistic model to calculate observation probabilities (scores). **However**, we wish to clarify that in our specific experimental setup, we utilize fully observed (complete) datasets to serve as ground truth for evaluation purposes. Consequently, this temporary imputation step is not utilized in our study. The logistic models calculate missingness probabilities directly from the complete observed values, ensuring precise control over the MNAR mechanism without the need for intermediate approximations.

---

> ### Author Response · Authors · 2025-12-20
> **Response (Part 3)**
>
> * [**C3**] The paper acknowledges that default hyperparameters were used for many baselines, but this may introduce bias. The authors should clarify which methods were most sensitive to hyperparameters, justify why deep learning models were not tuned, and discuss whether performance rankings might shift with tuning.
>
> **Response**: We thank the reviewer for raising this critical point regarding hyperparameter sensitivity and fair comparison. We have addressed this concern through three dimensions: computational feasibility, identifying sensitive methods, and empirical verification of ranking stability.
>
> **1. Computational Constraints and “Out-of-the-box” Evaluation**: Given the scale of our benchmark (3 datasets × 14 models × 3 mechanisms × 5 ratios × 5 random seeds ≈ 3,150 distinct experiments), performing a comprehensive grid search for every deep learning model would require over 30,000 runs, which is computationally prohibitive. Furthermore, a primary goal of this benchmark is to evaluate “out-of-the-box” performance, which reflects the practical reality for many researchers who rely on standard implementations. We followed the protocol of similar large-scale benchmarks$^{[1]}$,by using the official default configurations. Crucially, while we fixed hyperparameters, we ensured rigorous optimization: for all deep learning models, we implemented early stopping or set sufficiently long training epochs and inspected training loss curves to guarantee that models reached convergence, ruling out under-fitting as a cause for performance differences.
>
> **2. Identification of Sensitive Methods (MOT & MIRACLE)**: We acknowledge that certain methods are inherently more sensitive to hyperparameters. Specifically, we identify **MOT** (sensitive to the entropy regularization parameter $\epsilon$) and **MIRACLE** (sensitive to regularization weights $\beta _ 1$ and $\beta _2$) as highly volatile without tuning. We have added a specific discussion on this in the new **“Failure Analysis: MIRACLE and MOT”** subsection in **Section 6.2**, detailing how this sensitivity contributes to their failure modes.
>
> **3. Case Study on Ranking Stability (The GAIN Experiment)**: To scientifically address whether rankings shift with tuning, we selected **GAIN** (known for GAN instability) on the SubSDIC dataset (MNAR 30%) as a testbed.
>
> - **Convergence verification**: **We extended training from 1,000 to 10,000 epochs**. As shown in **Appendix A.8.3 (Figure 8)**, the loss stabilizes well before 1,000 epochs, confirming that default performance is not due to under-fitting.
>
> - **Grid search**: We tuned `hint_rate` ∈ [0.1, …, 0.9] and `loss_alpha` ∈ [20, …, 100]. The optimal configuration improved RMSE from **1.50 (default)** to **1.20 (tuned)**.
>
> - **Conclusion**: While tuning improved GAIN’s absolute performance (raising its rank by 3 positions), it remained significantly outperformed by the top-tier methods (MissForest, DSAN, ReMasker, DiffPuter), which all achieved RMSE scores **below 1.0**.
>
> This suggests that while tuning helps, the relative superiority of the top-tier methods is robust and primarily driven by model architecture rather than hyperparameter choices. We have added these details to **Appendix A.8.3**. The experimental results can be found in the supplementary material folder named `Experiment_result/SubSDIC/Imputation_performance/Out-sample`.
>
> ### Reference:
> - [1] [Jäger, S., Allhorn, A. and Bießmann, F., 2021. A benchmark for data imputation methods. Frontiers in big Data, 4, p.693674.](https://www.frontiersin.org/journals/big-data/articles/10.3389/fdata.2021.693674/full)

---

> ### Author Response · Authors · 2025-12-20
> **Response (Part 4)**
>
> * [**C4**] Discuss failure modes of specific models providing short diagnostic section to help contextualize the benchmark.
>
> **Response**: We thank the reviewer for this suggestion. Including a diagnostic analysis of failure cases provides crucial context for the results. We have added a new subsection “**Failure Analysis: MIRACLE and MOT**” in Section 6.2.
>
> Specifically, we attribute the poor performance and occasional convergence failure of **MIRACLE** and **MOT** to the following factors:
>
> - **MIRACLE:**
>   - **Hyperparameter sensitivity**: MIRACLE optimizes a complex objective combining reconstruction loss with causal ($𝓡 _ {causal}$) and moment ($𝓡 _ {moment}$) regularizers. As observed in prior benchmarks $^{[1]}$, this method is highly sensitive to hyperparameters ($\beta _ 1$, $\beta _2$). Without extensive dataset-specific tuning, the optimization often diverges, leading to the “empty results” or extremely high errors we observed.
>
>   - **Bootstrap divergence**: MIRACLE iteratively refines a baseline imputation. In our socioeconomic datasets, simple baselines (such as Mean) yield high initial errors due to the missingness generation mechanism for MAR and MNAR (see Appendix A.6). This poor initialization can cause MIRACLE to learn incorrect causal graphs, trapping the refinement loop in a local optimum or causing divergence.
>
>   - **Assumption violation (MNAR)**: Under MNAR, our mechanism involves self-masking. This explicitly violates MIRACLE’s Assumption 3 (no self-masking missingness), rendering the moment regularizer mathematically invalid and causing estimation collapse.
>
> - **MOT:**
>   - **Numerical instability**: MOT relies on the Sinkhorn algorithm to approximate optimal transport. This algorithm is known to be numerically unstable when the entropy regularization parameter ($\epsilon $) is not perfectly tuned. In our survey datasets, the standard heuristic for ε often fails, leading to numerical overflows or underflows.
>
>   - **Geometric limitation**: MOT minimizes distances in the original feature space. As noted by Zhao et al. (2023) $^{[1]}$, the Euclidean distance often fails to capture the complex underlying manifold of structured survey data, matching samples that are spatially close but semantically distinct.
>
> ---
>
> * [**C6**] A slightly more detailed explanation of why models like GRAPE or IGRM are incompatible with the evaluation framework would improve clarity.
>
> **Response**: We thank the reviewer for the helpful suggestion. We agree that our original explanation for excluding GRAPE and IGRM was too brief, and providing additional detail improves the clarity of the paper. These graph-based methods, in their public implementations, perform imputation in the graph-embedding space and compute reconstruction metrics within that latent space. Our evaluation framework requires feature-wise imputations in the original tabular feature space to support downstream training and comparison. Making these models compatible would require adding an additional layer that maps graph embeddings back to continuous and categorical variables, which introduces extra architectural choices and hyperparameters, thereby breaking comparability under our fixed compute budget.
>
> We have updated the Limitations section of **Section 7 (Conclusion)** to provide a clearer and more detailed explanation. We appreciate the reviewer’s suggestion, which has indeed improved the clarity of the manuscript. Future work can develop prediction models based on heterogeneous graphs, which might be better suited to the structural properties of socioeconomic survey data.
>
> ### Reference
> - [1] [Zhao, H., Sun, K., Dezfouli, A. and Bonilla, E.V., 2023, July. Transformed distribution matching for missing value imputation. In International Conference on Machine Learning (pp. 42159-42186). PMLR.](https://proceedings.mlr.press/v202/zhao23h.html?ref=https://coder.social)

---

> ### Author Response · Authors · 2025-12-20
> **Response (Part 5)**
>
> * [**C5**] The paper acknowledges that default hyperparameters were used for many baselines, but this may introduce bias. The authors should clarify which methods were most sensitive to hyperparameters, justify why deep learning models were not tuned, and discuss whether performance rankings might shift with tuning.
> * Additional Comment  [**A1**]: How sensitive are results to the choice of downstream models? RF/XGB are robust, but some imputation methods that leverage nonlinear interactions may perform differently under linear models.
>
> **Response**: We thank the reviewer for these insightful comments and for prompting us to clarify and extend our downstream evaluation.
>
> **Choice of RF and XGBoost.**
> In our experiments, the downstream tasks are standard classification and regression problems with a moderate number of features, for which classical machine-learning models are well suited. Our goal is to evaluate imputation quality under downstream models that are simple and widely used, rather than assuming that researchers or practitioners facing these problems would deploy large, heavily parameterized deep networks. In typical machine-learning practice, ensembles are often preferred over single models, and there are two dominant ensemble paradigms: **bagging** and **boosting**. Random Forest and XGBoost serve as canonical and widely adopted representatives of these two families, respectively. They therefore provide a reasonable and practically relevant choice of downstream models for our benchmark.
>
> **Hyperparameters for downstream models.**
> Downstream models were not tuned beyond reasonable defaults. We use standard library settings, with minimal adjustments such as setting `class_weight="balanced"` for Random Forest to account for label imbalance. The purpose of our benchmark’s downstream performance evaluation is threefold: (1) to compare 14 different imputers under the *same* downstream model; (2) to assess whether the relative ordering of imputers is stable across downstream models; and (3) to check whether classification and regression tasks induce different rankings. None of these goals requires optimizing each downstream model to its best possible performance, so we believe that well-motivated defaults are sufficient and avoid conflating imputation quality with extensive model tuning. We now clarify this in the revised text of **Section 6.3** Downstream Task Performance.
>
> **Additional downstream models and a “no-imputation” baseline.**
> To enrich the evaluation and directly address the reviewer’s suggestions, we extended the downstream analysis on the public SubSDIC dataset, which is the only dataset in our benchmark that naturally supports both a classification and a regression task. For each task, we additionally evaluate two types of downstream models:
>
> (a) **Linear models:** logistic regression for classification and linear regression for regression.
>
> (b) **LightGBM:** evaluated in two modes:
> &nbsp;&nbsp;&nbsp;&nbsp;1. using imputed data, and
> &nbsp;&nbsp;&nbsp;&nbsp;2. using LightGBM’s native handling of missing values as a naïve “no-imputation” baseline.
>
> The LightGBM experiments show that using the raw, non-imputed data with native missing-value handling yields the *worst* performance across all missingness mechanisms and most missingness ratios, for both classification and regression (see **Figure 10** in the revised manuscript). Even the weakest imputer in our pool, when combined with LightGBM, outperforms the no-imputation baseline. This supports the practical recommendation that, in our setting, explicit imputation is preferable to relying solely on LightGBM’s built-in missingness handling. We have included this analysis in **Appendix A.9.2**.
>
> **Sensitivity to the choice of downstream model.**
> To specifically address the concern that “some imputation methods that leverage nonlinear interactions may perform differently under linear models,” we compare the rankings of all 14 imputers across four downstream models: Logistic Regression, Random Forest, XGBoost, and LightGBM for classification; and Linear Regression, Random Forest, XGBoost, and LightGBM for regression. We visualize these rankings via bump charts (see Figure 6 in the revised manuscript) and quantify consistency using Kendall’s W defined in Eq. (3). **For classification, Kendall’s W=0.91, and for regression, Kendall’s W=0.95**. Although individual methods may move by one or two positions, the overall ordering is highly consistent across linear and nonlinear downstream models. This indicates that our main conclusions about the relative performance of imputers are not sensitive to the particular choice of downstream model.
>
> We have **added a brief description of this ranking consistency to Section 5.3.2 and provide the detailed result analysis and figures in Section 6.3**. We appreciate the reviewer’s suggestions, which have led to a clearer justification of our design choices and a more comprehensive downstream evaluation.

---

> ### Author Response · Authors · 2025-12-20
> **Response (Part 6)**
>
> * Additional Comment [**A2**]: Could the benchmark be extended to assess imputation uncertainty? Several deep generative models produce uncertainty estimates and incorporating proper scoring rules could open an interesting dimension.
>
> **Response**: We thank the reviewer for this forward-looking suggestion. We fully agree that uncertainty quantification is a critical frontier for modern imputation research, especially with the rise of probabilistic generative models. However, for this initial benchmark, we prioritized point estimation accuracy and downstream predictive utility. This scoping decision was driven by two key factors: first, many widely used baselines provide deterministic point estimates. Second, in the context of socioeconomic data analysis, practitioners frequently rely on single-imputed datasets for standard regression/classification, making a robust point estimation benchmark essential. We acknowledge uncertainty quantification as a valuable direction for future expansion; our released dataset, which includes complete ground truth, is ideally suited for evaluating uncertainty quantification metrics, and we encourage the community to utilize it for extending the benchmark into the probabilistic dimension.
>
> ---
>
> * Additional Comment [**A3**]: Do the authors plan to provide a leaderboard extension of this benchmark?
>
> **Response**: Yes, absolutely. We recognize that a dynamic leaderboard is essential for the long-term vitality of a benchmark. Accordingly, we have revised the Future Work section in the **Section 7 (conclusion)** to explicitly state our plan to release a leaderboard on PapersWithCode.
>
> To maintain the integrity of the double-blind review process, we have not yet launched the public leaderboard, as PapersWithCode require linking to non-anonymized repositories or arXiv preprints. However, we are fully prepared to establish the official leaderboard on **PapersWithCode** immediately upon publication.
>
> We will initialize the leaderboard with the detailed performance metrics of all 14 baselines evaluated in this paper, creating an immediate reference point for the community.

---

### Review · Reviewer_rUAa · 2025-11-29

**Summary Of Contributions:**

Disclaimers: I am not a specialist in missing data, but I am familiar with machine learning benchmarks and empirical methodology; my review focuses on the soundness and usefulness of the benchmark rather than fine-grained statistical theory.

This paper presents a comprehensive benchmark of missing-data imputation methods for large-scale socioeconomic survey data. The authors construct three datasets: (i) a proprietary real panel survey (CPHS) from India, (ii) a high-fidelity synthetic version of CPHS (SynthCPHS), and (iii) a publicly shareable subset of the World Bank’s SDIC synthetic dataset (SubSDIC). They simulate missingness under MCAR, MAR, and MNAR mechanisms at five missingness ratios (10–50%), and evaluate 14 classical and modern imputation methods. Evaluation covers imputation accuracy on continuous and categorical variables, impact on downstream classification and regression tasks (via AUC degradation and RMSE increase), and computational efficiency. The results show that classical methods such as MissForest perform very well at low missingness, while recent deep methods like ReMasker, HyperImpute, DSAN/DSN, and DiffPuter become competitive or superior under higher missingness or more complex settings. The authors also examine ranking consistency across datasets and release SubSDIC plus an evaluation framework to facilitate future research.

**Audience:**

Yes

**Audience Explanation:**

For researchers who need large-scale socioeconomic survey data for their experiments, the paper is helpful on providing a benchmark. For researchers and practitioners who need to understand what are good algorithms and methods available for missing-data imputation scenarios, the benchmarks of 14 methods are helpful as evidences and advices.

**Claims And Evidence:**

Yes

**Claims Explanation:**

The paper is mainly about dataset construction and benchmarking, not about novel algorithms. I reviewed it in the following aspects, 1) whether the tasks are helpful 2) how the datasets are designed and constructed; 3) whether the benchmarking is thourough and reasonable; 4) whether the benchmarking reveals something helpful for future studies.

For these questions, I think
1) Missing-data imputation in large socioeconomic surveys is an important applied problem;
2) The combination of real CPHS, a synthetic CPHS-like dataset, and a publicly shareable SubSDIC dataset is well thought-out;
3) The benchmark includes 14 methods spanning simple baselines, classical iterative algorithms, distribution-matching approaches, and several recent deep and diffusion-based methods;
4) MissForest as a strong, efficient default at low missingness; deep models becoming preferable when missingness is severe; DSN outperforming DSAN in out-of-sample settings—that will be genuinely useful to applied researchers choosing an imputation method.

**Requested Changes:**

1. Clarify how a third party can reproduce at least a substantial subset of your experiments using only publicly available components;
2. Hyperparameter tuning protocol and fairness across methods. It seems that many hyperparameters are set as defaults.
3. Sharpen the practical recommendations for practitioners. The empirical results contain many useful patterns (e.g., MissForest / HyperImpute remaining strong and efficient at low missingness, deep models outperforming at high missingness). To make the paper more actionable for applied users, I recommend adding a short “Practical Recommendations” subsection.

---

> ### Author Response · Authors · 2025-12-20
> **Response (Part 1)**
>
> We appreciate the reviewer rUAa’s insightful comments and suggestions. We address each point below.
> * [**C1**] Clarify how a third party can reproduce at least a substantial subset of your experiments using only publicly available components
>
> **Response**:We place a high priority on reproducibility. Third parties can reproduce a substantial subset of our experiments, specifically the full suite of evaluations on the **SubSDIC** dataset, using the following components and steps:
>
> **1. Data Access**: The **SubSDIC** dataset is publicly available and has been included in our **supplementary material**. This dataset supports both classification and regression tasks.
>
> **2. Code and Workflow**: We provide a comprehensive, open-source codebase (currently in an anonymous repository, to be hosted on GitHub upon publication). Reproducing the experiments involves three simple steps:
>
> 1. **Setup**: Download the SubSDIC data from the supplementary material and the code from the repository.
>
> 2. **Configuration**: Refer to the detailed README file provided in the repository. Users only need to modify the file paths in the configuration section (explicitly marked with capitalized comments, e.g., `# NEED TO CHANGE THE PATH...`).
>
> 3. **Execution**: Run the main scripts following the tutorial to perform preprocessing, imputation training, and downstream evaluation.
>
> **3. Scope of Reproducibility**
> - **Datasets**: We explicitly clarify that reproducibility is limited to **SubSDIC**. The CPHS dataset is proprietary and commercially restricted. Similarly, while SynthCPHS is synthetic, it retains highly specific statistical properties of the original data and is subject to strict agreements that prevent public release. However, the code pipeline used for SubSDIC is identical to that used for CPHS, allowing verification of the methodology.
>
> - **Outputs**: Our provided pipeline generates all numerical results reported in the supplementary material. While the core pipeline focuses on generating these quantitative metrics, the raw results are saved in CSV files, allowing users to visualize the results using their preferred plotting tools.
>
> We have added this guideline and scope of reproducibility in **Appendix A.2**.
>
> ---
>
> * [**C2**] Hyperparameter tuning protocol and fairness across methods. It seems that many hyperparameters are set as defaults.
>
> **Response**: We thank the reviewer for raising this important question regarding the tuning protocol and fairness. We acknowledge that using default hyperparameters is a design choice that requires justification. We addressed this through a standardized protocol, convergence verification, and a sensitivity case study.
>
> **1. Tuning Protocol and Fairness**: Our protocol standardizes hyperparameters by using the **official default configurations** recommended by the original authors. This approach ensures fairness by preventing subjective bias, where researchers might inadvertently tune specific methods more aggressively than others. Furthermore, benchmarking “out-of-the-box” performance reflects the practical reality for practitioners. Comprehensive grid search was also computationally prohibitive: our benchmark involves 3 datasets × 14 models × 3 mechanisms × 5 ratios × 5 seeds ≈ 3,150 distinct experiments. Tuning every deep learning model would require more than 30k runs. Crucially, to ensure this protocol did not lead to unfair under-fitting, we implemented early stopping or set sufficiently long training epochs for all deep learning models, inspecting loss curves to guarantee convergence.
>
> **2. Identifying Limitations (MOT & MIRACLE)**: We frankly acknowledge that this protocol affects methods differently. We identified MOT (sensitive to $\epsilon $) and MIRACLE (sensitive to $\beta _ 1$, $\beta _2$) as highly volatile under default settings. We have explicitly discussed this limitation in the new **“Failure Analysis”** subsection in **Section 6.2**.
>
> **3. Empirical Verification of Ranking Stability**: To verify whether the lack of tuning unfairly penalized specific models, we conducted a case study on **GAIN** (known for its sensitivity) on the SubSDIC dataset (MNAR 30%).
>
> - **Convergence**: **We extended training to 10k epochs.** As shown in **Appendix A.8.3**, the loss stabilizes well before the default 1k epochs, confirming the fairness of the training duration.
>
> - **Grid search impact**: We tuned `hint_rate` and `loss_alpha`. The optimal configuration improved RMSE from 1.50 (default) to 1.20 (tuned).
>
> - **Conclusion**: While tuning improved GAIN’s absolute score, it remained significantly outperformed by top-tier methods (MissForest, DSAN, ReMasker, DiffPuter, all with RMSE < 1.0).
>
> This evidence suggests that while default settings may not extract peak performance for every model, the relative rankings of top-tier methods are robust, preserving the fairness of the benchmark’s core conclusions. Detailed results are added to **Appendix A.8.3** and the `Supplementary_Material`.

---

> ### Author Response · Authors · 2025-12-20
> **Response (Part 2)**
>
> * [**C3**] Sharpen the practical recommendations for practitioners. The empirical results contain many useful patterns (e.g., MissForest / HyperImpute remaining strong and efficient at low missingness, deep models outperforming at high missingness). To make the paper more actionable for applied users, I recommend adding a short “Practical Recommendations” subsection.
>
> **Response**: We thank the reviewer for this excellent suggestion and agree that distilling the empirical results into actionable advice significantly enhances the paper's utility. As requested, we have added a new subsection, **Section 6.7: Practical Recommendations**, which synthesizes our findings into a hierarchical selection strategy. This section explicitly guides practitioners to use **MissForest** for an optimal balance of efficiency and robustness in low-to-medium missingness scenarios ($\leq 30\\%$), and to switch to specific deep learning methods (such as **ReMasker**, **DiffPuter** and **DSAN**) when performance is the first priority in high-missingness scenarios ($\ge 40\\%$). This addition addresses the trade-offs between computational cost and performance, making the benchmark results immediately actionable for applied users.

---

### Review · Reviewer_bSNe · 2025-12-10

**Summary Of Contributions:**

This submission addresses the critical gap in benchmarking missing data imputation methods for socioeconomic surveys, a domain characterized by longitudinal, hierarchical, high-dimensional, and non-i.i.d. data with complex missing mechanisms. The key contributions include: (1) a comprehensive benchmark integrating three datasets (real CPHS, synthetic SynthCPHS, and public SubSDIC) that reflect real-world socioeconomic survey characteristics; (2) systematic evaluation of 14 imputation methods across three missingness mechanisms (MCAR, MAR, MNAR) and five missingness ratios (10%-50%), covering both continuous and categorical variables; (3) multi-metric assessment encompassing imputation accuracy, downstream task performance (classification and regression), and computational efficiency; and (4) provision of an open-source framework and public dataset (SubSDIC) to support reproducible research.


### Key Strengths
1. Fills a critical niche by focusing on socioeconomic surveys.
2. Uses both real and high-fidelity synthetic datasets, addressing privacy constraints of proprietary survey data.
3. Adopts a holistic evaluation approach, moving beyond imputation accuracy to include downstream task impact and computational efficiency.
4. Comprehensive coverage of missingness scenarios (including the understudied MNAR mechanism) and diverse imputation methods (statistical, iterative ML, deep learning, hybrid) ensures generalizability of findings.

### Weaknesses
1. Graph-based imputation methods are excluded without sufficient justification, limiting the scope of comparative analysis.
2. No statistical significance tests are conducted when comparing imputation methods, making it unclear whether performance differences are statistically meaningful or due to random variation.
3. Figure 3 shows a counterintuitive trend: the 'Mean' curve’s RMSE decreases as missingness ratios increase, which contradicts typical expectations and lacks sufficient explanation to support its credibility.
4. Table 1 lacks detailed descriptive text, including clear definitions of column headers and brief context for each dataset, hindering reader's understanding of the benchmark dataset comparisons.

**Audience:**

Yes

**Audience Explanation:**

Yes. TMLR’s audience includes researchers and practitioners in machine learning, statistics, and applied data science, all of whom stand to benefit from this work.
1. For ML researchers, the paper provides a rigorous benchmark for evaluating imputation methods on structured, real-world data.
2. For applied researchers and policymakers working with socioeconomic surveys, the findings offer actionable guidance on selecting imputation methods that balance accuracy, downstream utility, and efficiency.
3. The open-source framework and public SubSDIC dataset enable further research on missing data in high-stakes domains.

**Broader Impact Concerns:**

The paper includes a brief ethical consideration section (Appendix A.1) addressing privacy protections for SynthCPHS.

**Claims And Evidence:**

Yes

**Claims Explanation:**

Yes. The authors provide rigorous empirical evidence to support their claims:
1. Distribution similarity between CPHS and SynthCPHS is validated via KS test (for continuous variables) and JS divergence (for categorical variables), with negligible differences and high p-values.
2. Downstream task impact is assessed using ROC-AUC degradation and RMSE increase, with results replicated.
3. Computational efficiency is measured via wall-clock time, with clear distinctions between fast traditional methods and slower deep learning approaches.
4. Ranking consistency is verified using Kendall’s coefficient of concordance, demonstrating strong agreement across datasets for key metrics.

However, the lack of significance tests for method comparisons and unexplained trends in Figure 3 slightly weaken the robustness of the evidence.

**Requested Changes:**

1. Provide additional justification for excluding graph-based imputation methods. If their output is incompatible with the evaluation framework, explain how this incompatibility arises and whether it could be resolved (e.g., post-processing to align with tabular format).
2. Conduct statistical significance tests to compare performance differences between imputation methods. Report p-values and clarify which method comparisons are statistically significant, especially for top-performing methods.
3. Provide a detailed analysis of the counterintuitive trend in Figure 3.
4. Enhance Table 1 with comprehensive descriptive text.

---

> ### Author Response · Authors · 2025-12-20
> **Response (Part 1)**
>
> We are grateful for the reviewer bSNe's detailed feedback. Our responses are provided below, with comments numbered for clarity.
> * [**C1**] Provide additional justification for excluding graph-based imputation methods. If their output is incompatible with the evaluation framework, explain how this incompatibility arises and whether it could be resolved (e.g., post-processing to align with tabular format).
>
> **Response**: We thank the reviewer for raising this point and for asking us to clarify the exclusion of graph-based imputers. As we now explain more explicitly in the revised manuscript, methods such as GRAPE and IGRM, in their public implementations, perform imputation in a learned graph-embedding space and compute reconstruction losses directly in that latent space. Their outputs are node/edge embeddings rather than per-feature values. By contrast, our evaluation framework is defined in the original tabular feature space: we require column-wise imputations for continuous and categorical variables, and we train downstream models on these reconstructed features.
>
> In principle, one could attempt to “post-process” or adapt these methods by adding a decoder that maps graph embeddings back to the original feature space. However, this is not a simple formatting step: it would require specifying and training an additional prediction head (with separate architectures for continuous and categorical variables, regularization, etc.), introducing substantial new design choices and hyperparameters. This would effectively define a new variant of each method rather than faithfully evaluating the original algorithms, and it would also violate our fixed compute and tuning budget, making comparisons with the other baselines less fair.
>
> To avoid these confounding factors, we chose not to include GRAPE/IGRM in the current benchmark and instead treat graph-based imputers as a separate line of work. As noted in the updated Limitations section of **Section 7 (Conclusion)**, the future work can focus on prediction models based on heterogeneous graphs that might be better aligned with the structural properties of socioeconomic survey data; in those study, they can design a standardized graph-to-tabular decoding protocol and compare GRAPE, IGRM, and related methods under matched computational budgets. We believe this separation yields a clearer, more interpretable benchmark in the present paper.

---

> ### Author Response · Authors · 2025-12-20
> **Response (Part 2)**
>
> * [**C2**] Conduct statistical significance tests to compare performance differences between imputation methods. Report p-values and clarify which method comparisons are statistically significant, especially for top-performing methods.
>
> **Response**: We thank the reviewer for this suggestion. To ensure statistical rigor, we conducted pairwise t-tests (based on 5 random seeds) for all 14 methods on the **SubSDIC** dataset. This resulted in 91 pairwise combinations across 15 experimental settings (3 mechanisms × 5 ratios) and 2 task types (continuous and categorical imputation), yielding a total of 2,730 p-values. We found that **84%** of these comparisons showed statistically significant differences (p < 0.05).
>
> Based on the analysis in Section 6.7, we highlight the statistical verification of the top-performing methods (**MissForest, ReMasker, DiffPuter, DSAN, DSN, HyperImpute**) in four key scenarios:
>
> **For Continuous Variables (RMSE)**
>
> **1. Low Missingness (10%, MAR/MNAR)**: MissForest is significantly superior to the other top models (p < 0.05). This confirms its dominance in low-missingness regimes.
>
> | Scenario | Method A (Winner) | Method B (Comparison) | p-value | Significant? |
> |--------|-------------------|-----------------------|---------|--------------|
> | MAR 10% | MissForest | ReMasker | 0.0418 | Yes |
> | MAR 10% | MissForest | DSAN | 0.0346 | Yes |
> | MAR 10% | MissForest | DSN | 0.0044 | Yes |
> | MAR 10% | MissForest | DiffPuter | 0.0000 | Yes |
> | MAR 10% | MissForest | HyperImpute | 0.0001 | Yes |
> | MNAR 10% | MissForest | ReMasker | 0.0061 | Yes |
> | MNAR 10% | MissForest | DSAN | 0.0222 | Yes |
> | MNAR 10% | MissForest | DSN | 0.0338 | Yes |
> | MNAR 10% | MissForest | DiffPuter | 0.0089 | Yes |
> | MNAR 10% | MissForest | HyperImpute | 0.0015 | Yes |
>
> **2. High Missingness (50%, MAR/MNAR)**: DSN significantly outperforms other baselines (p < 0.05). However, the difference between DSN and DSAN is not statistically significant (p > 0.05), which aligns with our observation in **Figure 3** that their performance is comparable (DSN is a simplified version of DSAN without the attention layer).
>
> | Scenario | Method A (Winner) | Method B (Comparison) | p-value | Significant? |
> |--------|-------------------|-----------------------|---------|--------------|
> | MAR 50% | DSN | ReMasker | 0.0025 | Yes |
> | MAR 50% | DSN | MissForest | 0.0005 | Yes |
> | MAR 50% | DSN | DiffPuter | 0.0078 | Yes |
> | MAR 50% | DSN | HyperImpute | 0.0022 | Yes |
> | MNAR 50% | DSN | ReMasker | 0.0021 | Yes |
> | MNAR 50% | DSN | MissForest | 0.0086 | Yes |
> | MNAR 50% | DSN | DiffPuter | 0.0000 | Yes |
> | MNAR 50% | DSN | HyperImpute | 0.0069 | Yes |
>
> **For Categorical Variables (F1 Score)**: We omit the table here; detailed results can be found in the supplementary material.
>
> 1. **Low-to-Medium Missingness (≤ 20%, All Mechanisms)**: MissForest significantly outperforms all 13 other methods (p < 0.05), demonstrating exceptional robustness in this scenario.
>
> 2. **High Missingness (50%)**: ReMasker significantly outperforms other methods (p < 0.05), with the exception of DiffPuter. The difference between ReMasker and DiffPuter is not statistically significant, consistent with Figure 4, where both generative models show similar high performance at extreme missingness ratios.
>
> We have included the full set of 2,730 p-values in the `Supplementary_Material`. We also added a reference to these tests in the conclusion of Section 6.7. Finally, we created a new **Appendix A.9.1: Statistical Significance Analysis**, which details the methodology and summarizes these key comparisons.

---

> ### Author Response · Authors · 2025-12-20
> **Response (Part 3)**
>
> * [**C3**] Provide a detailed analysis of the counterintuitive trend in Figure 3.
>
> **Response**: We thank the reviewer for pointing out this counterintuitive trend. We acknowledge that for Mean Imputation, a decreasing RMSE under increasing missingness seems counterintuitive. However, we have empirically verified that this is a correct and expected characteristic of the logistic missingness generation mechanism used in our benchmark. We have updated the end of **Section 4.3** to clarify that this mechanism simulates the “tail censoring” effect (e.g., income non-response), which is supported by survey methodology literature $^{[1, 2]}$ as a realistic feature of socioeconomic data.
>
> To validate this experimentally, we quantified the “extremeness” of the missing values by calculating the average absolute Z-score of the ground truth values that were masked. A detailed analysis of this phenomenon, including the methodology and the validation plot (**Figure 8**), is now provided in the newly added **Appendix A.6 (Analysis of Missing Value Extremeness and RMSE Trends)**. As shown in Figure 8, under MAR and MNAR mechanisms:
>
> - **At low missingness ratios (10%)**: The logistic mechanism is highly selective, masking primarily the outliers (observations furthest from the mean). Since Mean Imputation replaces these extreme values with the global mean, the residuals are maximized, resulting in a high RMSE.
>
> - **At high missingness ratios (50%)**: The mechanism becomes less selective. To reach the 50% target, it must mask a large portion of data points lying closer to the mean. While Mean Imputation still fails on outliers, it performs reasonably well on these newly added points lying closer to the global mean, thus lowering the average error.
>
> In contrast, the MCAR curve in our analysis remains flat, confirming that this trend is specific to value-dependent masking. We have revised the manuscript to explicitly explain this phenomenon in **Section 6.2 (Imputation Performance)**.
>
> ### Reference
> - [1] [Riphahn, R.T. and Serfling, O., 2005. Item non-response on income and wealth questions. Empirical economics, 30(2), pp.521-538.](https://link.springer.com/article/10.1007/s00181-005-0247-7)
> - [2] [Meyer, B.D., Mok, W.K. and Sullivan, J.X., 2015. Household surveys in crisis. Journal of Economic Perspectives, 29(4), pp.199-226.](https://www.aeaweb.org/articles?id=10.1257/jep.29.4.199)
>
> ---
>
> * [**C4**] Enhance Table 1 with comprehensive descriptive text.
>
> **Response**: We thank the reviewer for pointing this out. We agree that the previous version of Table 1 was too concise. To improve clarity and context, we have comprehensively revised the table and its caption in **Table 1**. We expanded the caption to explicitly define all column headers. To preserve the readability of the main text, we have included the detailed background context and source links for each of the eight existing datasets in **Appendix A.4**. This ensures readers have access to the full dataset descriptions without overcrowding the main comparison table.

---

### Decision · Action_Editor_zZYq · 2026-01-12

**Recommendation:** Accept as is

**Additional Comments:**

The paper presents a solid evaluation of a diverse selection of imputation techniques and how these affect downstream machine learning tasks. The reviewers uniformly supported the acceptance of this paper into TMLR.

**Audience:**

Yes

**Audience Explanation:**

Imputation is one of the most common preprocessing steps. Identifying strengths and weaknesses of popular methods of value.

**Claims And Evidence:**

Yes

**Claims Explanation:**

The paper constructs a solid benchmark evaluation of a representative choice of imputation techniques.